# Time-Series-Based Queries on Stable Transportation Networks Equipped with Sensors

Erik Bollen [1,2,*], Rik Hendrix [2], Bart Kuijpers [1] and Alejandro Vaisman [3]

1.  Databases and Theoretical Computer Science Group and Data Science Institute (DSI), Hasselt University, 3590 Diepenbeek, Belgium; bart.kuijpers@uhasselt.be
2.  Flemish Institute for Technological Research (VITO), Data Science Hub, 2400 Mol, Belgium; rik.hendrix@vito.be
3.  Department of Information Engineering, Instituto Tecnológico de Buenos Aires, Buenos Aires 1245, Argentina; avaisman@itba.edu.ar
*   Correspondence: erik.bollen@uhasselt.be

**Abstract:** In this paper, we propose a formalism to query transportation networks that are equipped with sensors that produce time-series data. The core of the proposed query mechanism is a logic-based language that is capable to return time, value, and time-series outputs, as well as Boolean queries. We can also use the language for node selection and path selection. Furthermore, we propose an implementation of this language in a graph database system and evaluate its working on a fragment of the Flemish river system that is equipped with sensors that measure the water height at regular moments in time.

**Keywords:** time-series data; transportation networks; sensors; constraint databases; logic-based query languages

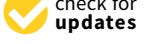



## 1. Introduction and Motivation

A *sensor network* [1] is a collection of sensors that monitor data in different locations, and send these data to a data center for storage, viewing, and analysis. There are many applications for sensor networks, from monitoring a single home, to the surveillance of a large city, to earthquake detection for the whole world [2]. Furthermore, researchers, farmers, and governments need to monitor aspects of the natural environment such as air pollution, water quality, soil conditions, and weather metrics. For example, two recent studies concerning soil can be found in [3,4]. In lakes, for example, buoys use sensors to collect data about wind speed, water temperature, air temperature, and wave height. Further, wireless sensor networks are also used in small-scale environmental monitoring, as they can be implemented usually at an affordable cost for fine-grained environmental sensing and monitoring. Intelligent transportation systems [5] are increasingly being used in a wide range of applications, from traffic monitoring, wildlife tracking, habitat monitoring, among others. Both ITSs and sensor networks are currently very active fields of research. This paper addresses both fields together. These ideas can be extended to any kind of transportation network. We study transportation networks equipped with sensors that produce time-series data. Further, both the transportation networks and the sensors are taken in a broad sense such that a wide range of applications are covered. We illustrate what we intend with a series of examples. Our first examples come from the field of (vehicle-based) transportation [6], and here, a transportation network could be a road network on which sensors are placed (for example, as cameras above the roads or induction loops in the roads) that measure the traffic density. Yet another example is an urban heating system through which heated water is transported [7], or a power grid through which electricity is transported [8]. On such networks, we can imagine sensors that measure the water temperature or pressure, or the voltage of the electrical current. Another relevant

example of a transportation network, which is the one that is addressed in this paper, is the case of a river system on which sensors are placed to measure, at regular moments in time, the water height, the water temperature, the pH, or salinity (or conductivity) of the water. Because of the relevance of the problem of water monitoring, the Flemish Institute for Technological Research in Belgium (VITO) plans a large deployment of the sensors on the rivers in Flanders, with, among others, the aim of building models and facilitating inter-sensor validation. This project is called the "Internet of Water" [9].

A characteristic that all the above examples have in common is that the underlying transportation network is rather *stable*, in the sense that the changes in the network over time are minimal. Roads may be closed, rivers may be dammed, and a power line may fail, but these events occur only occasionally. In this paper, we consider transportation networks that are spatially well defined and stable in the above sense. Examples of networks that are unstable include friendship networks on social media, and examples of networks that are spatially vague include migration routes of animals over the globe, and we do not address them here. The interested reader can find examples related to social networks in Cozza et al. [10].

Most of the existing work in the field of transportation networks carrying sensors that produce time-series data focuses on reducing the data traffic in the network and the associated energy consumption, for example, by pre-aggregating time-series data produced by sensors in these networks, or producing samples of these series [11,12]. There is also work addressing the selection of optimal location of the sensor nodes in a network [13]. The general approach, particularly in road traffic prediction, is based on neural networks, deep learning, and Artificial Intelligence techniques, combined with time series analysis. This is comprehensively reviewed in [6]. Nevertheless, there is not much work addressing both the network topology and the time series associated with the nodes [14].

In this paper, we consider the problem of *querying and managing data related to a sensor-equipped stable transportation network*. We assume that the data produced by a sensor are a sequence of (time,value)-pairs on which the natural ordering on time induces an ordering. As we discuss in the related work section, there exists work specifically to deal with time-series data, extensively surveyed by Jensen et al. [15]. There is also existing work concerning query languages for time series, where SQL is extended to deal with ordered data (see, e.g., in [16]), but this does not include the second dimension of our problem either, namely, the network topology. In summary, a number of proposals exist to query time-series data, but none of these is particularly geared towards the geographical setting of transportation networks that we have described above.

As mentioned, most of the work in the field of ITS is based on applying different techniques, like data mining and machine learning techniques or time-series analysis, to predict parameters of interest at some location in the network, ignoring the network topology. Our approach is different. We address the problem from a spatio-temporal database point of view, where time series *and* network topology are first-class citizens. In this sense, we consider the sensor-equipped network as a graph, whose nodes and edges are associated with time-series data. Based on this structure, a high-level query language can be devised, to express queries like "Give me the nodes in the river network such that the water height has been constantly increasing in the last ten hours", or, further, to prevent floods, we could ask "Give me the nodes in the river network, located downstream of nodes where the water height has been constantly increasing in the last ten hours." This cannot be done with existing proposals.

In light of the above, the main contribution of this paper is a formal query language for transportation networks equipped with sensors that produce time-series data. The core of the proposed query mechanism builds on a logic-based language (or calculus) that works on both the network connectivity and the time-series data. This language is capable of returning time, value, and time-series outputs, as well as Boolean values. We can also use the language for node selection and path selection. As it is customary in the field of databases, we present our query language using a logic-based formulation [17]. As

technically, our query language belongs to the field of of (linear) constraint databases [18,19], a natural way to proceed to obtain a working version of this language would be to turn to an existing constraint database implementation, such as DEDALE [20,21], DISCO [22], or MLPQ [23]. We consider this as future work, but as a proof of concept, we instead propose an implementation of our language over a graph database system [24], and evaluate it on a fragment of the Flemish river system that is equipped with sensors that measure the water height at regular moments in time. Graph databases have gained in popularity during the past decade and they are specifically suited to model transportation networks, which are at the basis of sensor networks. In this approach, we use the Neo4j graph database (http://neo4j.com (accessed on 2 October 2020)), one of the most widely used graph databases in the marketplace, which comes equipped with a high-level query language, Cypher [25]. In our approach, Cypher is extended with functions that allow handling time series. Time series can be added to the graph either as a list of (time,value)-pairs, or as a connection to a time-series-specific database, like InfluxDB (https://www.influxdata.com/ (accessed on 10 December 2020)). This approach falls within the tradition of Object-Relational database systems, that extend relational databases with functions allowing to address new paradigms [26]. We show in this use case how the formal query language is used as the basis for expressing queries over the implemented high-level query language.

The remainder of this paper is organized as follows. In Section 2, we give an overview of related work on transportation networks and time-series data management. In Section 3, we define the data model for transportation networks equipped with sensors that produce time-series data, on which the formal query language is built. This language is presented in Section 4, along with a large number of examples and a characterization of the different classes of queries the language can express. In Section 5, we present the use case example of the Flanders river system, which is represented as a graph and stored in a Neo4j graph database (Neo4j). A collection of representative queries is expressed in the formal query language, and then implemented in the Cypher extension mentioned above. The paper concludes in Section 6, with a discussion on this work and the open research challenges.

## 2. Related Work

Literature on sensor networks is extremely vast on both foundations and applications for different real-world problems. Regarding the former, Akyilidiz et al. [1] survey sensor networks, their characteristics, and architectures, and discuss design factors like scalability, production costs, fault tolerance, topology, and power consumption. The foundations for sensor networks are described in the work by Delin and Jackson [27], who study how several different sensors can be combined into a network and the benefits of such a combination for analysis purposes. This idea is extended by Ventura et al. [28], who devise an implementation of an heterogeneous network of sensors. Further, Cannate et al. [29] present an implementation of a sensor observation system for the Ticino canton in Switzerland. None of these works include the graph component, which we address in the present paper. The protocol stack of a sensor network is presented as a set of layers, namely, the physical, data, network, transport, and application layers. Our work positions in the latter. Sensor networks address different kinds of transportation problems. For example, Karami and Kashef [6], present data, methods, and models for intelligent transportation planning, in particular for traffic forecasting. In the same context, Sanchez et al. [30] show that analyzing and predicting traffic data using machine learning is possible. They use a graph neural network to predict the traffic at intersections, where the flow from one intersection to another is represented by a graph. As mentioned in Section 1, these algorithms are based on data mining, machine learning, and time-series analysis techniques, such as in the work of Peng et al. [31]. The same ideas can be extended to other kinds of transportation networks, like energy grids, where data science methods are being massively applied. The application of data science methods in energy networks is comprehensively surveyed by Zhang et al. [8]. The authors characterize the big data problem on smart grids based on big data's famous five V's: volume, velocity, variety, veracity, and value. We finally

mention that this topic is strongly related to the so-called Internet of Things (IoT) [32], as discussed in [33], where the authors remark that the IoT sensor data consist of complex time-series data, and thus require efficient data analysis mechanisms. These mechanisms include data aggregation and artificial intelligence techniques. Previous work of the authors of the present paper compares the use of relational databases and graph databases to store and query network topologies, also using the Flanders river system as a case study [34]. That work addresses typical topological queries (e.g., shortest path, shortest distance, and transitive closure) over the river system. However, none of the work above addresses the problem of querying, in a database style, the network topology together with time-series data.

Time-Series Management Systems (TSMS) are extensively surveyed by Jensen et al. [15]. The survey covers existing systems and prototypes, and classifies them in three categories: (1) Internal stores, (2) Extended stores, and (3) Relational Database Management System Extensions. Features considered include purpose of the systems, whether or not it can be distributed, the maturity of the system, and if the system provides a high-level query language. As this is relevant for the work presented here, we remark that only nine out of twenty-seven systems have an associated query language. Further, three of these are based on relational databases. In almost all cases the query language is an SQL-like variant. There are few time-series specific query languages. Among these, Lerner and Sasha propose AQuery [16], an SQL-based query language that takes the order of records into account, which is appropriate for time-series data. Nevertheless, AQuery is not used in any of the surveyed systems in [15]. Second, AQuery accounts for time series, but it does not consider other features like, for instance, spatial relationships between the time series. Sadri et al. [35] propose SQL-TS, an SQL extension which, similarly to the AQuery language, also processes a sequence of data. Actually, the main difference with AQuery is that SQL-TS is based on the relational model. The main focus of SQL-TS is achieving fast pattern matching on a sequence of data, which differs from the goal of the present paper. Another work concerning this temporal query language topic is the paper of Seshadri et al. [36] propose SEQ, a database system supporting sequence. SEQ includes a declarative sequence query language called SEQUIN, based on an algebra of query operators, which allows query optimization and evaluation. SEQ is a component of the PREDATOR database system that supports relational and other kinds of complex data [37]. Recently, extensive work dealing with temporal data at a lower level has been researched by Kvet et al. [38].

Property graphs [39] extend graphs with the capability of annotating nodes and edges with properties. Over this model, most graph databases [24,40] are built. Angles et al. [41] study the foundations of query languages for property graph databases. Property graph databases are used in the present paper to represent and store the transportation networks. Sensor data can be stored as properties of nodes and/or edges. These properties are time series which can, in turn, be represented either as a sequence of (time,value)-pairs, or as a pointer to a database where the time-series data resides. As mentioned in Section 1, one of the most popular graph databases in the marketplace is Neo4j, which is used in the case study discussed in the present paper. Neo4j comes equipped with a high-level query language, denoted Cypher. The formal semantics of the read-only portion of Cypher is studied by Francis et al. [25]. As a follow-up of that work, Green et al. [42] also study to address updates over Cypher. This semantics is captured by the part of the language proposed in Section 4 of this paper. In the proof of concept of Section 5, Cypher is extended with functions that implement the time-series part of the formal language, and the binding between the nodes and the time series. Furthermore, in Section 5, we compare and link the formal query language and the corresponding Cypher implementation.

## 3. Transportation Networks Equipped with Sensors That Produce Time-Series Data

In this section, we give the definition of a transportation network that is equipped with sensors that produce time-series data. We start with an example that illustrates how we model such networks. Figure 1 shows, on the left-hand side, a river system that consists

of thirteen river segments. The blue arrows indicate the flow direction of the water and the start and end locations of the river segments are indicated by blue circles. The segments are numbered 1 to 13 and this numbering is used in the directed graph on the right-hand side of Figure 1, where segments are represented as nodes, and the directed edges (shown in red) represent the FlowsTo relationship. A pair of nodes $(i, j)$ in the FlowsTo relationship indicates that water flows (directly) from river segment $i$ into river segment $j$. We refer to the graph on the right part of Figure 1, as a transportation network (in this example, of the river system).

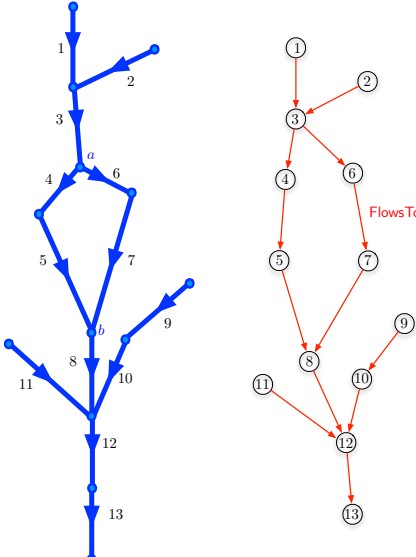

**Figure 1.** A fragment of a river system is shown on the left-hand side. The river is represented as a topology consisting of different segments. The numbered river segments carry arrows that indicate the flow of the water. On the right-hand side, the corresponding transportation network is shown. Here, the river segments are represented as nodes and the edges indicate the direction of the flow of the water. In other words, the arrows represent the FlowsTo relation.

We remark that the river system could also be modeled representing junctions or particular points as nodes, and segments as edges between two nodes such that, in this case, water flows from one node to another one. Both representations are equivalent and the former one is chosen in this work [34].

We note that in the river system of Figure 1, the river splits at the end of segment 3 (at location $a$) and the two branches come together again at location $b$, where we have the confluence of the river segments 5 and 7.

We remark that, in general, the transportation network of a river system is acyclic, as naturally flowing water cannot flow from one location via some path to that same location (if the river system includes pumps, this might be different). On the other hand, the transportation network of a road system typically allows for cycles.

We are now ready to give the definition of a transportation network.

**Definition 1.** *A transportation network* $\mathcal{TN}$ *is a directed graph* $(N, E)$*, where $N$ is a finite set of nodes and $E \subseteq N \times N$ is a set of directed edges.*

The graph on the right-hand side of Figure 1 is an example of a transportation network with $N = \{1, 2, ..., 13\}$ and $E$ is the FlowsTo relation (as indicated by the red arrows).

As a notational convention, we use the expressions $n, n', n_1, n_2, ...$ for variables that range over the set of nodes in a transportation network, and we use sans-serif letters $\mathsf{n}, \mathsf{n}', \mathsf{n}_1, \mathsf{n}_2, ...$ to refer to constant nodes in a transportation network. For the edge relation, we use the binary predicate $E(\mathsf{n}_i, \mathsf{n}_j)$ to express that there is an edge from $\mathsf{n}_i$ to $\mathsf{n}_j$. As we

assume the relation $E(n_i, n_j)$ to be present in this way, we will not require variables and constants for edges.

In our examples, we focus on river systems and we call the transitive closure of the FlowsTo relation, the DownStream relation.

Next, we give the definition of a sensor-equipped transportation network. In this definition, we need a set $\mathbb{T}$ of (possible) time moments and a set $\mathbb{V}$ of (possible) measurement values. We assume that both $\mathbb{T}$ and $\mathbb{V}$ are ordered sets.

**Definition 2.** *A sensor-equipped transportation network (or* sensor network, *for short* SN*) is a 4-tuple* $(N, E, S, \mathsf{TS})$, *such that* $(N, E)$ *is a transportation network,* $S \subseteq N$ *is a set of sensor-equipped nodes* (sensors, *for short), and* $\mathsf{TS} : S \to \mathbb{V}^{\mathbb{T}}$ *is a (*time-series*) function that maps sensors to a finite function from* $\mathbb{T}$ *to* $\mathbb{V}$.

With $\mathbb{V}^{\mathbb{T}}$, we denote the set of functions from $\mathbb{T}$ to $\mathbb{V}$. We remark that a finite function from $\mathbb{T}$ to $\mathbb{V}$ will be a partial function, in general. In practical applications, we will usually have $\mathbb{T} = \mathbb{R}$ and $\mathbb{V} = \mathbb{R}$, but the sets $\mathbb{T}$ and $\mathbb{V}$ can also be finite. We assume that these sets are at least equipped with a (total) order relation (denoted $\leq$), but they may also be equipped with functions (such as $+$ and $\times$). We remark that the order on the set $\mathbb{T}$ induces a natural order on the time series. As the function $\mathsf{TS}$ produces a finite function from $\mathbb{T}$ to $\mathbb{V}$, this means that no time moment can be associated with more than one measurement value in a time series. Finally, we remark that a time series is modeled as a set, rather than a linked list. The total order on $\mathbb{T}$ induces a natural order on such a set.

As a notational convention, we use $t, t', t_1, t_2, \ldots$ for variables that range the time set $\mathbb{T}$ and we use sans-serif letters $\mathsf{t}, \mathsf{t}', \mathsf{t}_1, \mathsf{t}_2, \ldots$ to refer to constant time moments. Similarly, $v, v', v_1, v_2, \ldots$ are used for variables that range the value set $\mathbb{V}$ and we use sans-serif letters $\mathsf{v}, \mathsf{v}', \mathsf{v}_1, \mathsf{t}_2, \ldots$ to indicate constant measurement or sensor values. Furthermore, we use $S$ as a predicate on the set $N$, which returns true on nodes that are sensors.

Figure 2 shows a sensor network, based on the transportation network of Figure 1. In this example, $S = \{4, 11\}$ and their times series are $TS(4) = \{(1, 2.5), (2, 3.5), (7, 4.5), (11, 9.5)\}$ and $TS(11) = \{(2, 17.7), (9, 3.5), (17, 4.5), (111, 19.5)\}$. We use this example to emphasize that sensors, in the case of river systems, are associated with river segments (in this example, with segments 4 and 11). This means that the exact geographical location of the sensor is not specified as such, which might pose a problem when geographical queries come into play. Our definition also allows for only one sensor per segment. However, these are shortcomings that are easily remedied and we discuss them in Section 6.

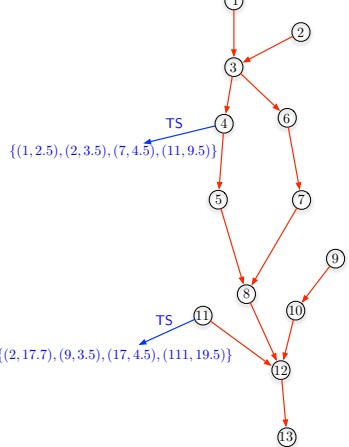

**Figure 2.** An example of a sensor network with the time series attached to the sensor nodes 4 and 11 indicated in blue. Again, we note that the nodes represent the segments of a river.

## 4. A Formal Language for Querying Transportation Networks Equipped with Sensors

In this section, we give a proposal for a logic-based language to query transportation networks in which some nodes are equipped with sensors that produce time-series data.

At the basis of our query language is a first-order logic language, which we call *time-series calculus* and which we denote by $\mathcal{TSC}$. We first define this basic language in Section 4.1. In Section 4.2, we illustrate the capabilities of $\mathcal{TSC}$ to return time, value, and time-series outputs and also show how $\mathcal{TSC}$ can be used for node selection and path selection. Next, we describe more powerful extensions of $\mathcal{TSC}$ in Section 4.3.

### 4.1. The Time-Series Calculus $\mathcal{TSC}$ to Query Sensor Networks

In this section, we describe the query language $\mathcal{TSC}$ on sensor networks. We assume that a set of predicates $\mathcal{P}_{\mathsf{TN}}$ on the transportation network is given. To $\mathcal{P}_{\mathsf{TN}}$ belong binary predicates like the edge-relation $E$ (which is FlowsTo, when we work on river systems) and the unary relation $S$ (to select sensor nodes). However, depending on the application, many more predicates may belong to $\mathcal{P}_{\mathsf{TN}}$, for example, the transitive closure of the edge relation (which is the DownStream relation, when we work with river systems).

Now, we describe the logic-based query language $\mathcal{TSC}$, which we call the *time-series calculus*. The language $\mathcal{TSC}$ aims at expressing queries on possibly several time series in a sensor network to

- produce a value in $\mathbb{T}$ or $\mathbb{V}$,
- produce a new time series,
- give a Boolean answer,
- etc.

The language $\mathcal{TSC}$ uses three types of variables: node variables, time variables, and value variables. These variables are used to talk about nodes in the transportation network, moments in time, and values that are measured by the sensors, respectively. In the following definition and thereafter, we follow the notational conventions of Section 3 for variables and constants. For generality, we assume, from now on, that $\mathbb{T} = \mathbb{R}$ and $\mathbb{V} = \mathbb{R}$, as this will be sufficient for most applications.

**Definition 3.** *A time-series calculus formula $\varphi$ is a first-order logic formula built using the connectives $\neg, \wedge$ and quantification ($\exists, \forall$) over time variables, value variables and node variables starting from the following atomic formulas:*

- *$(t, v) \in \mathsf{TS}(n)$;*
- *$T \leq 0$, where $T$ is a linear polynomial in some time variables $t_1, .., t_m$ and some value variables $v_1, .., v_k$; and*
- *$P(n_1', ..., n_\ell')$, for a $\ell$-ary predicate $P \in \mathcal{P}_{\mathsf{TN}}$.*

*Such a formula can have a number of free time-variables, value-variables, and node-variables, and to show this dependence, we use a notation like $\varphi(t_1, ..., t_\alpha, v_1, ..., v_\beta, n_1, ..., n_k)$, where $\alpha, \beta$ and $k$ are natural numbers (which are possibly 0).*

For reasons of finite representability, we will require that constants that are used as coefficients in the linear polynomials in this definition have some finite encoding. For example, we might say that a constant should be a rational, algebraic, or Turing-computable number.

We remark that $\leq$ suffices to express $=$ and $<$ in the obvious way: $T = T' := T \leq T' \wedge T \leq T'$ and $T < T : T \leq T \neg (T = T')$, for two linear polynomials $T$ and $T'$.

To illustrate the use of the calculus $\mathcal{TSC}$ and motivate the use of linear polynomial inequalities, we give some examples.

**Example 1.** *To express that the values in the time series of a sensor equipped node n are increasing, we would need to write that for any two time-value couples $(t, v)$ and $(t', v')$ in the time series of*

$n$, we have that $t \leq t'$ implies that $v \leq v'$. In our language $\mathcal{TSC}$, we would formally write this by the formula

$$\forall t \forall t' \forall v \forall v'(((t,v) \in \mathsf{TS}(n) \wedge (t',v') \in \mathsf{TS}(n)) \wedge t - t' \leq 0) \rightarrow v - v' \leq 0.$$

In this example, we use the linear polynomials $t - t'$ and $v - v'$.

**Example 2.** *To express that the measurements in a time series of node $n$ are at least 5 units (assuming that our unit on time is minutes, for example) apart, we would need to express that for consecutive time-value tuples $(t,v)$ and $(t',v')$ in the time series of $n$, we have that $t' \geq t + 5$.*

*To write the above condition as a tsc formula, we first need to define a predicate $NextTime(t,t',n)$ that expresses that in sensor node $n$, we have measurements at time $t$ and $t'$, but not at any moment in between them. This could be written as*

$$NextTime(t,t',n) := \exists v \exists v'((t,v) \in \mathsf{TS}(n) \wedge (t',v') \in \mathsf{TS}(n) \wedge$$
$$\neg(\exists t'' \exists v''((t'',v'') \in \mathsf{TS}(n) \wedge (t < t'' < t')))).$$

*Then, the formula*

$$\forall t \forall t'(NextTime(t,t',n) \rightarrow t + 5 - t' \leq 0)$$

*expresses that the measurements in a time series of node $n$ are at least 5 units apart in time (using the linear polynomial $t + 5 - t'$).*

**Example 3.** *As a final example to illustrate the use of linear polynomials, we consider the query that expresses that the speed of increase in the measured values is at least 2. We could express this by writing that*

$$\frac{\Delta v}{\Delta t} \geq 2 \quad or \quad \frac{v' - v}{t' - t} \geq 2$$

*for consecutive time-value tuples $(t,v)$ and $(t',v')$ in the time series of $n$. This amounts to the formula*

$$\forall t \forall t' \forall v \forall v'(((t,v) \in \mathsf{TS}(n) \wedge (t',v') \in \mathsf{TS}(n)) \wedge NextTime(t,t',n)) \rightarrow 2(t' - t) - (v' - v) \leq 0.$$

*In this example, we used the linear polynomial $2(t' - t) - (v' - v)$ which uses time- and value-variables in a mixed way.*

Before we explain the semantics of the query language $\mathcal{TSC}$, we remark that we intend node variables in $\mathcal{TSC}$ formulas to range over the nodes in the input sensor network, whereas we intend time and value variables to range over $\mathbb{R}$ (or $\mathbb{T}$ and $\mathbb{V}$, in general). This means that we do not intend them to be restricted to values that appear somewhere in the time series in the sensor network that is the input to the query. In this sense, our proposed language $\mathcal{TSC}$ can be seen to belong to the framework of a *(linear) constraint databases* [18,19].

The following definition specifies the semantics of queries expressed by time-series calculus formulas.

**Definition 4.** *A time-series calculus formula $\varphi(t_1, ..., t_\alpha, v_1, ..., v_\beta, n_1, ..., n_k)$, when evaluated on a sensor network $\mathcal{SN} = (S, N, E, \mathsf{TS})$, defines a set*

$$\{(t_1, ..., t_\alpha, v_1, ..., v_\beta, n_1, ..., n_k) \in \mathbb{T}^\alpha \times \mathbb{V}^\beta \times N^k \mid \mathcal{SN} \models \varphi[t_1, ..., t_\alpha, v_1, ..., v_\beta, n_1, ..., n_k]\},$$

*in the sense of first-order constraint database logics [18,19]. This set is considered the output $Q_\varphi(\mathcal{SN})$ of the query $Q_\varphi$, defined by the formula $\varphi(t_1, ..., t_\alpha, v_1, ..., v_\beta, n_1, ..., n_k)$, when evaluated on input the sensor network $\mathcal{SN}$.*

We remark that, as in the previous definition (and as is usual in logic), we use square brackets, to indicate instantiations of free variables in a formula by constants. For example, if $\varphi(t, v, n)$ is a formula with the three indicated free variables, and we want to instantiate $t$ by $\mathsf{t}_0$ and $n$ by $\mathsf{n}_1$, we would write $\varphi(t, v, n)[t = \mathsf{t}_0, n = \mathsf{n}_1]$. We also simply write $\varphi(\mathsf{t}_0, v, \mathsf{n}_1)$ when it is necessary to contain the length of the expressions, for example.

Now, we show that the output of $\mathcal{TSC}$-queries is computable (on input a sensor network).

**Proposition 1.** *Queries expressible by time-series calculus formulas can be effectively evaluated.*

Below, we sketch the proof and refer to previous work on constraint databases for the details [18,19].

**Proof.** When we want to evaluate a formula $\varphi(t_1, ..., t_\alpha, v_1, ..., v_\beta, n_1, ..., n_k)$ on a sensor network $\mathcal{SN} = (S, N, E, \mathsf{TS})$, we need to consider a finite number of possibilities for time- and value-variables that appear in expressions of the form $(t, v) \in \mathsf{TS}(n)$, as the number of nodes in a network is finite and the number of entries in any time series is also finite. This means that $\exists$-quantification on those variables can be replaced by a finite disjunction and $\forall$-quantification on these variables can be replaced by a finite conjunction. For each of these finitely many possibilities, there are other time- and value-variables to be considered and they can take infinitely many values (when $\mathbb{T}$ and $\mathbb{V}$ are $\mathbb{R}$). However, the resulting formula falls within the formalism of constraint databases with linear polynomials constraints only and query evaluation is known to be effective in this context [18,19]. $\square$

### 4.2. Example Queries

In this section, we illustrate the capabilities of the calculus $\mathcal{TSC}$ to return (sets of) time moments, values, time-series, and nodes. Furthermore, we also show how it can define paths and return Boolean answers.

### 4.2.1. Time- and Value-Queries

Queries that return (a set of) time moments or values are expressed by $\mathcal{TSC}$-formulas with $\alpha = 1, \beta = 0$ and $\alpha = 0, \beta = 1$ (as given in Definition 3), respectively.

**Definition 5.** *A $k$-ary time-query is expressed by a time-series calculus formula of the form $\varphi(t, n_1, ..., n_k)$. A $k$-ary value-query is expressed by a time-series calculus formula of the form $\varphi(v, n_1, ..., n_k)$.*

The formulas $\varphi$, that we are interested in here, have, apart from a number of free node-variables $n_1, ..., n_k$ (to denote the dependence on the time series of these nodes), additionally a free time-variable or a free value-variable. If $\varphi(t, n_1, ..., n_k)$ has only the additional free time-variable $t$ or $\varphi(v, n_1, ..., n_k)$ has only the additional free time-variable $v$, the output of the query is a set of time moments or a set of values (depending on the nodes $n_1, ..., n_k$), respectively.

Now, we give examples of time- and value-queries.

**Example 4.** *The $\mathcal{TSC}$-expression*

$$LastTime(t, n) := \exists v((t, v) \in \mathsf{TS}(n) \wedge \forall t' \forall v'((t', v') \in \mathsf{TS}(n) \rightarrow t' \leq t))$$

*defines the latest measurement moment $t$ for node $n$. This is an example of a formula that expresses a time-query. The formula $LastTime(t, n)$ produces a (singleton) subset of $\mathbb{T}$ (for a given node $n$). When we evaluate $LastTime(t, n)$ on the sensor network of Figure 2 in node $\mathsf{n} = 4$, we obtain the (set with the) time moment 11.*

**Example 5.** *As an example of a value-query that returns a value (depending on a node n), we consider "Return the last value of sensor n." The last value of sensor n is given by*

$$LastValue(v, n) := \exists t(LastTime(t, n) \wedge (t, v) \in \mathsf{TS}(n)).$$

*The formula $LastValue(v, n)$ produces a subset of $\mathbb{V}$ (for a node n). When we evaluate $LastTime(t, n)$ on the sensor network of Figure 2 in node $\mathsf{n} = 4$, we obtain the (set with the) value 9.5.*

We remark that, in the above examples, we failed to express that $n$ is a sensor node. This can be remedied by adding $S(n)$ to the formulas, but we assume that $(t, v) \in \mathsf{TS}(n)$ evaluates to false, whenever $n$ is not a sensor.

As a variation on $k$-ary time-queries and $k$-ary value-queries, we also discuss the case $\alpha > 0, \beta = 0$ and $\alpha = 0, \beta > 0$ by giving the following example.

**Example 6.** *We consider "Return the next measurement time $t'$ of sensor n after measurement time $t$." Consecutive moments $t$ and $t'$ for sensor n can be expressed by the tsc-formula $NextTime(t, t', n)$ of Example 2. When we evaluate $NextTime(t, t', n)$ on the sensor network of Figure 2 in node $\mathsf{n} = 4$, for $\mathsf{t} = 2$, we see that $NextTime(2, 7, 4)$ evaluates to true, indicating that $\mathsf{t}' = 7$ is the next measurement moment after $t = 4$ for this node.*

### 4.2.2. Time-Series Queries

If $\varphi(t, v, n_1, ..., n_k)$ has an additional free time-variable $t$ and an additional free value-variable $v$, the output of the query is a time series (when the defined set is finite). Here, we have $\alpha = \beta = 1$ using the notation of Definition 3.

**Definition 6.** *A $k$-ary time-series query is expressed by a time-series calculus formula of the form $\varphi(t, v, n_1, ..., n_k)$.*

We give an example of such a query.

**Example 7.** *As an example of a query that returns a time series, we consider the query "Return the subseries of the time series of node n consisting of values larger than 10." This query is expressed by the formula*

$$Larger10(t, v, n) := (t, v) \in \mathsf{TS}(n) \wedge 10 \leq v.$$

*For a node $\mathsf{n}$, it evaluates to true on all couples $(\mathsf{t}, \mathsf{v})$ in the time series of $\mathsf{n}$ for which $10 \leq \mathsf{v}$. We remark that the result of this query is a set which inherits an ordering from the total order on $\mathbb{T}$. When we evaluate $Larger10(t, n)$ on the sensor network of Figure 2 in node $\mathsf{n} = 11$, we obtain the set $\{(2, 17.7), (111, 19.5)\}$.*

We remark that the output defined by a formula $\varphi(t, v, n_1, ..., n_k)$ may not satisfy the definition of a time series. For example

$$\varphi(t, v, n) := \forall t' \forall v'((t', v') \in \mathsf{TS}(n) \rightarrow t \neq t') \wedge v = 0,$$

defines an infinite set (when $\mathbb{T} = \mathbb{R}$). This output is not a time series as time series are required to be finite. However, a *safety* condition could be added by expressing that the output of the query consists of isolated points in the set $\mathbb{T} \times \mathbb{V} = \mathbb{R}^2$ [18].

### 4.2.3. Node Selection Queries

In this section, we describe how Boolean conditions on time series are used as conditions to select nodes or tuples of nodes. We consider $\mathcal{TSC}$-formulas with $\alpha = \beta = 0$, using the notation of Definition 3.

**Definition 7.** *A k-ary node selection query is expressed by a time-series calculus formula of the form $\varphi(n_1, ..., n_k)$.*

When evaluated on a sensor network, the formula $\varphi(n_1, ..., n_k)$ defines an output which is a set of *k*-tuples of network nodes.

Now, we give examples of 1-ary and 2-ary node selection queries.

**Example 8.** *As an example of a unary node selection query, consider the query "Nodes n such that all the values in their associated time series are decreasing."*

*This query defines nodes based on their own time series and it is expressed by the formula*

$$Decrease(n) := \forall t \forall t' \forall v \forall v' \big( ((t, v) \in \mathsf{TS}(n) \wedge (t', v') \in \mathsf{TS}(n) \wedge t \leq t') \rightarrow v \geq v' \big).$$

*When we evaluate Decrease(n) on the sensor network of Figure* 2*, we obtain the empty set of nodes, as none of the sensor nodes have a decreasing time series. Similarly, we can write a formula*

$$Increase(n) := \forall t \forall t' \forall v \forall v' \big( ((t, v) \in \mathsf{TS}(n) \wedge (t', v') \in \mathsf{TS}(n) \wedge t \leq t') \rightarrow v \leq v' \big),$$

*which expresses that the values in a time series (of a node n) are increasing. This query, when evaluated on the sensor network of Figure* 2*, would return the node set $\{4\}$.*

**Example 9.** *As an example of a binary node selection query we consider the question "Give all node pairs $(n, n')$ such that $n'$ is reachable from $n$ by at most two edges, and such that the values in both their time series are decreasing." This query defines node pairs and it can be expressed, using the earlier expression as an abbreviation, as*

$$Down2Decrease(n, n') := (E(n, n') \vee \exists n''(E(n, n'') \wedge E(n'', n'))) \wedge Decrease(n) \wedge Decrease(n').$$

*When we evaluate $Down2Decrease(n, n')$ and $DownDecrease1(n)$ on the sensor network of Figure* 2*, we obtain the empty set as a result.*

4.2.4. Boolean Queries

Here, we consider $\mathcal{TSC}$-formulas without free variables, that is, $\alpha = \beta = k = 0$, using the notation of Definition 3.

**Definition 8.** *A Boolean query is expressed by a time-series calculus formula of the form $\varphi$ (without free variables).*

Based on the query examples of the previous section, we give the following example.

**Example 10.** *We consider the example "The values in the time series of all node n in the sensor network are decreasing." Obviously, this query can be expressed by the $\mathcal{TSC}$-formula*

$$AllDecrease := \forall n \ Decrease(n).$$

*When we evaluate AllDecrease on the sensor network of Figure* 2*, we obtain the value false, as the time series of node* 11 *is non-decreasing.*

4.2.5. Path Selection Queries

Path selection queries return paths in some form. Obviously, the proposed query language can only return sets (of nodes, time moments and value, and combinations thereof). Therefore, we need to assume some kind of encoding of paths in order to view a set of nodes as a path.

*Using binary node selection queries to express path selection queries.* When the underlying transportation network of a sensor network is a directed (multi-root) tree, like the network in Figure 2, paths can be represented by their start and end nodes. Indeed, in such networks,

there is either no path between two nodes or there is a unique path between them. In such a context, binary node selection queries can be used to output paths.

*Using unary node selection queries to express path selection queries.* However, unary node selection queries can also be used to return paths. Suppose that the formula $\varphi(n)$, when evaluated on a sensor network $\mathcal{SN} = (S, N, E, \mathsf{TS})$, returns a set of nodes $N_\varphi \subseteq N$. Then, we can write a "safety" query $\sigma_\varphi$ that expresses that $N_\varphi$ forms a path in $\mathcal{SN}$. This gives the following property.

**Proposition 2.** *If the formula $\varphi(n)$ expresses a unary node selection query, then the formula*

$$\varphi(n) \wedge \sigma_\varphi$$

*expresses a path selection query.*

We refer the reader interested in the detail to the Appendix A.

*4.3. Extensions of the Time-Series Calculus for Sensor Networks*

First-order logic, as a query language, has several known shortcomings. For example, it lacks the capability to express recursive queries [17,43]. On the other hand, it cannot express several aggregation queries that are needed if we want to perform Online Analytical Processing (OLAP) (this is explained below). There are several possibilities to overcome these issues and, in this section, we discuss several of them.

The problem of the inexpressibility of recursive queries manifests itself, in our context, mostly at the level of the transportation network, in the sense that we cannot express reachability queries. On the other hand, for our purposes, aggregation queries are most useful at the level of the time-series data.

In Section 4.3.1, we discuss reachability queries at the level of the transportation network and in Section 4.3.2, we discuss aggregation queries on subseries.

4.3.1. Reachability Queries

In Example 9, we defined pairs of nodes that were separated by one or two edges and that satisfied some condition (they both had decreasing time series). Similarly, we can write queries that involve nodes that are at most three, four, five, ... edges apart. However, writing queries about nodes that are an unspecified number of edges apart, would require the transitive closure $E^*$ of the edge relation $E$. On the level of queries that concern connectivity of nodes in a transportation network, it is well known that graph connectivity is not first-order expressible [17,43]. The following example focuses on river systems, where the transitive closure of the FlowsTo relation is called DownStream (see Section 3). We already remark that the query in the following example selects nodes based on a condition that involves the time series of downstream nodes.

**Example 11.** *As an example of a query that uses the transitive closure of a binary relation, we consider the binary node selection query that asks "Give all node pairs $(n, n')$ such that the values in a time series of the node n are decreasing and the values in all downstream nodes $n''$ are also decreasing and $n'$ is downstream of n." This query defines node pairs based on their own time series and those of all downstream nodes.*

*Using the earlier expression of Example 8 as an abbreviation, we can express this second query as*

$DownDecrease2(n, n') := \mathsf{DownStream}(n, n') \wedge$
$$Decrease(n) \wedge \forall n''(\mathsf{DownStream}(n, n'') \rightarrow Decrease(n'')).$$

*From this query, the unary node selection query "The values in a time series (of a node n) are decreasing and the values in all downstream nodes $n'$ are also decreasing" can be derived. It is expressed by the formula*

$$DownDecrease1(n) := Decrease(n) \wedge \forall n'(\mathsf{DownStream}(n, n') \rightarrow Decrease(n')).$$

*When we evaluate DownDecrease2(n, n') and DownDecrease1(n) on the sensor network of Figure 2 we obtain the empty set as a result, in both cases.*

Therefore, an easy solution to the reachability issue is to add the transitive closure of the edge relation to the set of predicates $\mathcal{P}_{\mathsf{TN}}$ (see Definition 3), that we use in the definition of $\mathcal{TSC}$. For our purposes this addition will be sufficient.

A more powerful generalization of this approach is to add a *transitive closure operator* in general. We mean that for any binary node selection query, expressed by a formula $\varphi(n_1, n_2)$, we allow, as an addition to $\mathcal{TSC}$, the expression

$$\mathsf{TC}[\varphi(n_1, n_2)].$$

For what concerns the semantics of such an expression, we define that, when $\varphi(n_1, n_2)$ evaluated on a sensor network $\mathcal{SN}$ gives the binary node relation $R_\varphi$, $\mathsf{TC}[\varphi(n_1, n_2)]$ evaluates to the transitive closure $R_\varphi^*$ of $R_\varphi$, when evaluated on $\mathcal{SN}$. Obviously, the transitive closure of the edge relation, $E^*$, is then obtained by the expression $\mathsf{TC}[E(n_1, n_2)]$. A transitive closure operator can be added in an even more general way, by allowing

$$\mathsf{TC}[\varphi(n_1, n_2, ..., n_k, n'_1, n'_2, ..., n'_k)]$$

on $2k$-ary relations. For further details on the semantics and termination properties of transitive closure logics in the context of constraint databases, we refer to the works in [44,45].

Another way to add this kind of expressive power to $\mathcal{TSC}$ is to consider a Datalog extension of $\mathcal{TSC}$ [17,43]. Languages with a recursion mechanism (such as Datalog) have already been studied extensively in the context of constraint database languages for dealing with spatial data. We refer to the works in [18,19,46] for more details.

### 4.3.2. Aggregation Primitives on Time Series

In our setting, it is often useful to be able to express aggregation queries on time-series data. This need occurs when we have OLAP operations in mind. An example of such an OLAP query is "Give the average of the measured values during the month of January for all sensors." Recall that in Section 1 we mentioned that it is usual to aggregate time series values to reduce the size of the series.

In general, when $\varphi(t, v, n_1, ..., n_k)$ expresses a $k$-ary time-series query (see Section 4.2.2), then we want to be able to perform aggregations on the the resulting set of values. The aggregation operations that we have in mind include the following.

- SUM, returning the sum of the selected values (per node tuple);
- AVG, returning the average of the selected values (per node tuple);
- COUNT, returning the count of the selected values (per node tuple);
- MAX, returning the maximum of the selected values (per node tuple); and
- MIN, returning the minimum of the selected values (per node tuple).

The above aggregation primitives return values in the set $\mathbb{V}$, except COUNT which produces a natural number. These primitives take time series as input (possibly parameterized by node tuples), but they work (per node tuple) on the set of values that are in the time series under consideration (and thus, ignore the time component).

We give an example.

**Example 12.** *We return to the above example of the unary time-series query: "Give the average of the measured values during the month of January for all sensors." Hereto, we assume that January is given by the time interval $[t_{Jan}^-, t_{Jan}^+]$.*

*The $\mathcal{TSC}$-formula*

$$Jan(t, v, n) := t^-_{Jan} \leq t \leq t^+_{Jan} \wedge (t, v) \in \mathsf{TS}(n)$$

*defines, for each sensor node n, the part of its time series belonging to January.*
*Then,*

$$\mathsf{AVG}[Jan(t, v, n)]$$

*produces couples $(\bar{v}, n)$, where $\bar{v}$ is the average of the time series for January of node n.*

The above list of aggregation primitives can be extended as needed by the application at hand.

## 5. Use Case Study: the Flanders River System

As the query language $\mathcal{TSC}$ and its extensions belong to the field of constraint databases, a natural way to proceed to obtain a working version of this language would be to turn to an existing constraint database implementation, such as DEDALE [20,21], DISCO [22], or MLPQ [23], as we mentioned in the introduction. We consider this future work, but as a proof of concept, we propose an implementation of our language over a graph database system and evaluate its working on a fragment of the Flemish river system that is equipped with sensors that measure the water height at regular moments in time. This use case is based on the "Internet of Water" project (https://www.internetofwater.be (accessed on 4 February 2021)), where such a system could be used to deal with the gathered sensor data.

To demonstrate the applicability of the proposed query language, we consider a number of example queries, in the river use case context, and give their expression in (extensions of) $\mathcal{TSC}$ but also over a Neo4j (http://www.neo4j.com (accessed on 2 October 2020)) graph database system implementation (Data and code are available on the OSf platform: DOI 10.17605/OSF.IO/95TCX), using the Cypher query language mentioned in Section 2. We consider the following list of example queries:

- $Q_1$: What is the current status of the network?
- $Q_2$: Give the state of the network during the last hour.
- $Q_3$: What is the average of the measurements for sensor A during 15th March 2020?
- $Q_4$: What was the highest measured value in the network during January, at which node was that?
- $Q_5$: What are the last 10 values of the nearest upstream sensor for node E?
- $Q_6$: What are the upstream sensor values at the moment that location G has its max value during 15th March?
- $Q_7$: What is the average value in each series for all downstream sensors of a location F?
- $Q_8$: What is the time difference between the moments where sensor G and D reach their maximum during the last day of 2020?
- $Q_9$: Do all sensors on a path from J to A have all measurements below a threshold value $\tau$ during interval $l$?
- $Q_{10}$: Return the paths where the sensors on a path from J to A have all measurements below a threshold value $\tau$ during interval $l$?

These queries will be discussed in the next sections. The queries are shown in their (extended) $\mathcal{TSC}$-formulation and their implementation in the Neo4j language Cypher. We also discuss the results of those queries on an example region of a river system. However, first, we discuss the part of the river system and the time-series data.

### 5.1. River System and Time-Series Data

The part of the Flemish river system under consideration consists of the river Demer between the Flemish cities of Diest and Aarschot in Belgium. The actual river is shown in Figure 3. This stretch of river and its side arms have different measurement stations that are all equipped to measure the water height. In Figure 3, the locations of the water-height

sensors are indicated with the green dots. In Figure 4, an abstract version of the same part of the river system (including the side arms) and the sensors is depicted. A node in this case represents a certain part of the river. All nodes can thus be called river segments (or segments for short). If somewhere on that segment a measurement station is available, then there is also a water-height time series that can be linked to the node. In that case, the node is also said to be a sensor. These nodes, that are a segment and a sensor, are shown with a pink color in the graph of Figure 4. If a considerable stretch of river exists that does not have a measuring station, a green-colored node, (without a time series) is added. For example, the river stretch near Messelbroek is considered to be long and therefore two nodes, F and E, are taken to represent this part in the graph. There are also two side arms for this system, the short and longer one correspond to the rivers near Molenstede (node N) and Rillaar (node M, L K), respectively.

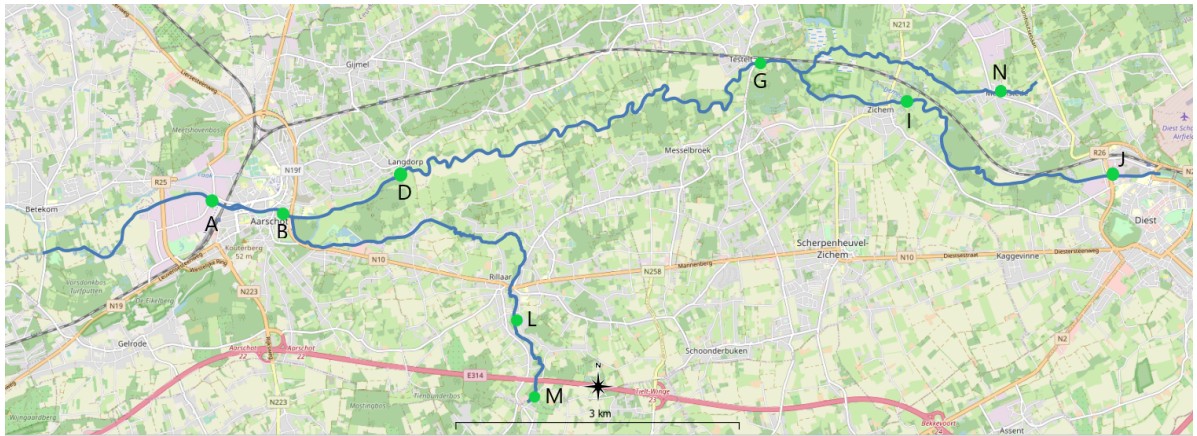

**Figure 3.** The part of the Flemish river system used in the experiments. The green dots indicate the locations of the sensors . The letters beside the dots are the IDs of such locations. This map is realized with QGis (`qgis.org`) and OpenStreetMap (`openstreetmap.org`).

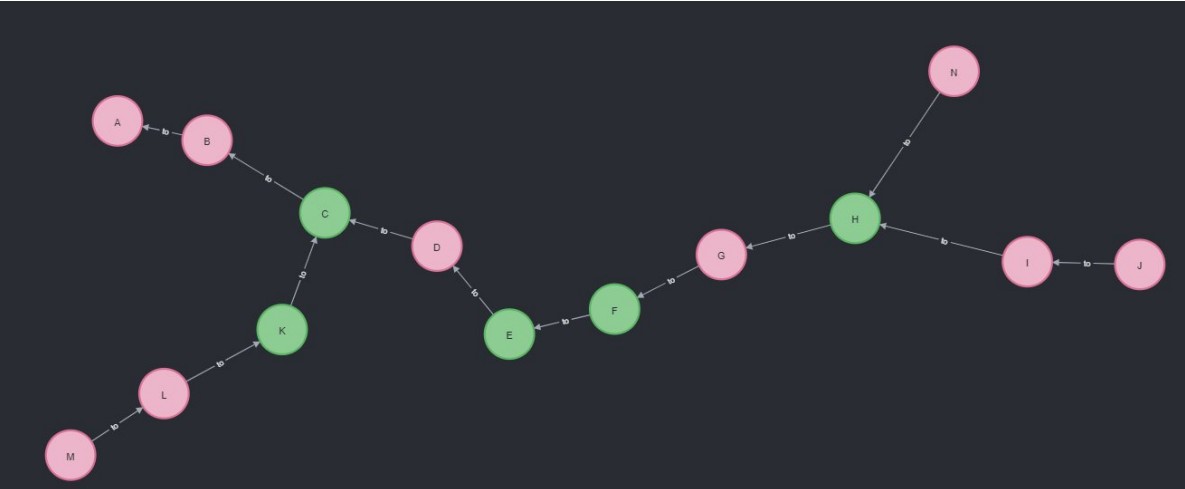

**Figure 4.** The transportation network version of the river system of Figure 3. All nodes correspond to river segments and the segments equipped with a sensor are indicated in pink. All segments have an ID which is a letter. For each segment with a sensor this ID coincides with the IDs used to denote the sensor in Figure 3.

The time-series data in this experiment are the water height measured by the stations. For each measuring station the data available on `waterinfo.be`, provided by VMM (https://en.vmm.be (accessed on 8 March 2021)) (Flemish Environmental Agency) are taken for the entire year of 2020. The data have a 15 min resolution. One time series thus has a maximum of 35,040 time-value pairs.

*5.2. Data Model and Cypher Extensions*

We first build the data model, where the spatial and temporal components are combined. This means, in practice, that time series need to be added to the nodes in the graph. Then, some extra functionalities need to be added to Neo4j that allow the user to add, delete, and query the newly created model.

As data structure in the data model, we define a list of key–value pairs, also known as a map. Adding a map to a node as a property effectively links the series to the node. However, adding a map to a node in Neo4j is not trivial. The default implementation allows properties to be key–value pairs where the key is a string (not starting with a number), and the value is of one of the supported default types or a list of one of these types. In addition, the value can not be map data structure. Therefore, the series is linked to the node by adding a property (*name*, *series*) to it, with *name*, in this case "level", being the name of the series and *series* a string of the map. By adding an extension to Neo4j that can interpret this property (by means of parsing *series* as JSON), the series is made accessible.

There are other possibilities to effectively link time-series data to graph data. We could, for example, store the graph and the time series in different systems. In this case, a Time-Series Management System (TSMS) could be used to store the time-series data and a graph database to store the network data. The advantage of this approach is that there exist specialized systems for those kinds of data, already equipped with appropriate functionalities. On the other hand, it is also possible to store all data in a graph. More specifically, in this approach the series are stored as a linked-list of nodes, where each node represents a measurement and thus has a time and a value attached to it. This linked list of measurement nodes is attached to a node of the network, and the nodes within the linked list are sorted on time, effectively placing the most recent measurement nearest to the network node. For these three models, discussed here, it is at the moment unclear whether or not one approach is better than the others. This is a question that can be investigated in future work. However, to demonstrate the working of the idea of $\mathcal{TSC}$, one alternative needs to be chosen. In this case, the object approach, as described in the previous paragraph, was chosen over a combination of TSMS and graph databases or the linked list graph-based approach. This solution is similar to the one taken for geometric data in relational databases, for example, in Postgres and PostGIS. In addition, the realization of this approach is more straightforward, because it does not require the use of multiple systems or advanced graph operations to select specific time–value pairs. However, this decision should not be seen as any evidence that may suggest that the approach is superior to the others. It is merely a design choice made in order to realize a proof of concept of the ideas presented in this paper.

In practice the following six functions or procedures are implemented for retrieving time-series data.

1. getValueDiscrete(node, name, timepoint)
2. getValueContinuous(node, name, timestamp)
3. getValuesDiscrete(node, name, List<timepoint>)
4. getValuesContinuous(node, name, List<timestamp>)
5. getValuesDiscreteRange(node, name, begin, until)
6. getValuesContinuousRange(node, name, begin, until)

The functions and procedures can be partitioned into two groups: a group of *discrete* and a group of *continuous* functions. The first group selects a value or timestamp–value pair based on a relative index with 0 being the most recent. The latter group selects the elements based on an absolute time. The functions that retrieve one value, *getValue...()*, are user functions that can be directly used in the Cypher queries and that evaluate itself into one value corresponding to the time-point or timestamp given. The other four functions are user-defined procedures, and they return a stream (can also be interpreted as a list) of timestamp–value pairs. These functions need to be called explicitly and the results need to be processed in the remaining Cypher query (For more information on procedures and working with them, we refer to https://neo4j.com/developer/cypher/procedures-

functions/ (accessed on 20 October 2020)). The two *getValues...()* procedures return a result stream based on the node, the series name, and a list of timestamps or time points. In contrast, the *getValues...Range()* procedures take a start time and end time and returns all timestamp-value pairs that fall within this range (including the *begin* and excluding *until*).

When using the continuous functions, an actual timestamp has to be supplied. However, it is not always possible to know the exact time of a reported value. If a sensor measures values on irregular timestamps, or with seconds or even milliseconds precision, supplying the exact time is nearly impossible. Requiring a measurement to be returned only if an exact timestamp is supplied may lead to invalid results. Therefore, it is of interest to implement the functions and procedures with a sort of *interpolation* functionality. In this case, we decided to take the following approach: a measurement is considered valid until a new measurement is available. This means that if a timestamp is given that does not correspond to an existing one in the database, the first timestamp back in time and its value are taken and returned. Even though a timestamp in the future may be closer to the supplied timestamp, it will not be considered. However, in other use cases and other implementations the latter option may be more adequate. This could even be extended to include advanced interpolation algorithms that take many parameters into account which, in turn, opens up an entire research field on its own. This all is also highly use case-dependent, and the implementation of a few generic algorithms is a foreseen topic in future work.

Furthermore, other functionalities are implemented, such as functions and procedures to add values to a specific time series or delete entries of it. However, these are not discussed in detail here, as they are not relevant when querying the time-series data. Similarly, nodes and edges in the network could be annotated with other additional data such as the geo-locations of the sensors. Although this is possible, we do not elaborate on this in order to keep the focus on the time-series aspect.

*5.3. Example Queries with Their Expression in $\mathcal{TSC}$ and Their Implementation in Cypher*

For each of the above example queries, a formal expression in $\mathcal{TSC}$ and its extensions is given along with an implementation in Cypher. For each query a predicate is defined that selects all tuples that correspond to the answer of the query.

**Q$_1$: What is the current status of the network?** (meaning: Give me all sensor nodes with their latest measurement.

We already discussed this query in Example 5 and we have $Q_1(v, n) := LastValue(v, n)$.

In Cypher, this query is expressed as follows:

```
1  MATCH (n:Series)
2  RETURN n.id, neo4j_tquery.getValueDiscrete(n,
3      "level" , 0);
```

Here, the implemented function returns the last value in the time series because for the discrete functions and procedures, the time series is treated as an array-like structure, where the last value is always at the begin of the array.

For this query, the result consists of 9 result records, each one consisting of an ID of the station and the last value. The record for the most downstream station is, for example, $(A, 9.75)$. An image of the actual output is shown in Figure 5.

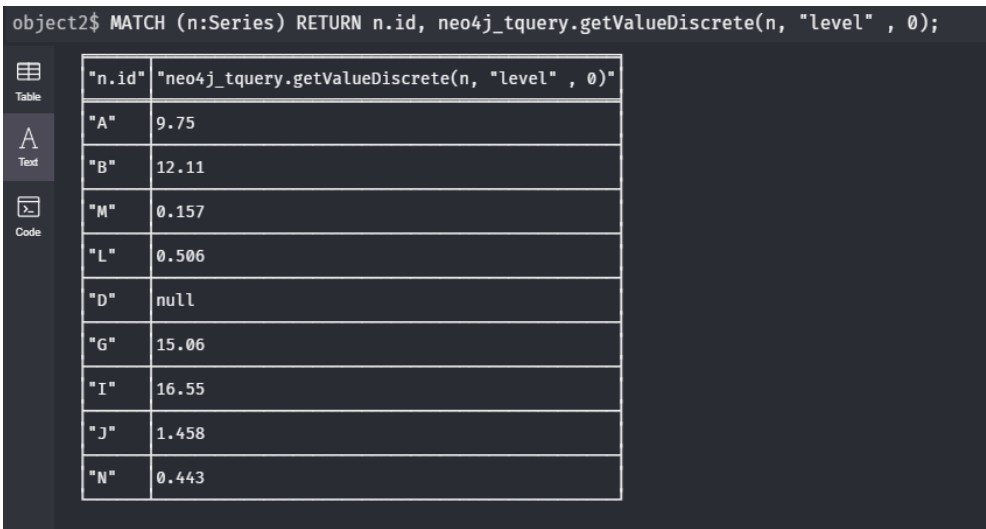

**Figure 5.** Visualization of the output of query $Q_1$ run on the sensor network of Figure 4. The result is taken from the web interface of Neo4j.

**Q$_2$: Give me the state of the network during the last hour?** (meaning: Give all sensor nodes with all their measurements during the last hour).

Let $t_{now}$ be the current time and let $t_h$ be a real number that corresponds to one hour. Then, we can express this query as $Q_2(t, v, n) := t_{now} - t_h \leq t \leq t_{now} \wedge (t, v) \in \mathsf{TS}(n)$.

In Cypher, this query is expressed as follows:

```
1  MATCH (n:Series)
2  CALL neo4j_tquery.getValuesDiscrete(n, "level" ,
3      [0,1,2,3])
4  YIELD timestamp AS t, value AS v
5  RETURN n.id, t, v;
```

Here, the last four values are requested because the measurements are taken every 15 min, and therefore we know that four measurements need to be requested. However, this information may not always be available to the user, therefore a more generic selection method is preferred. In query languages specially developed for time series, for example, Flux (https://docs.influxdata.com/influxdb/v2.0/query-data/get-started/ (accessed on 10 December 2020)), ranges can be created by supplying a timestamp and difference to that point. For example, $now - 1h$, meaning that the timestamp should be of the last hour. Our extension supports the keywords *now* and *epoch*, but these cannot be used together with an operator and constant value ($-1h$). However, this functionality can be realized in our proof of concept by using the *Duration* in Neo4j. First, a temporal type is created and this timestamp can be added or subtracted by a duration. The following query shows how this looks for the current query if it is run for the last hour of the first day of December.

```
1  MATCH (n:Series)
2  CALL neo4j_tquery.getValuesContinuousRange(n, "level",
3      left(toString(datetime("2020-12-01T23:59")
4          -duration({hours:1})), 19),
5      left(toString(datetime("2020-12-01T23:59")), 19))
6  YIELD timestamp AS t, value AS v
7  RETURN n.id, t, v;
```

We notice that the `ContiuousRange` procedure is used where the lower bound is the current time (datetime) subtracted with one hour and the upper bound is just the current time. The support for the keyword *now* is available in our extensions, but for the temporal types in Neo4j the *datetime()* function needs to be used. Moreover, the procedure expects timestamps as strings and without a timezone. In order to comply with this, the *toString()*

function is used after which the result is trimmed, with *left()*, by taking the first 19 characters of the string, effectively trimming the "Z" of the string. The output of the query is shown in Figure 6.

**Figure 6.** Visualization of the output of query $Q_2$ run on the sensor network of Figure 4. The result is taken from the web interface of Neo4j. Only a snippet of the entire result is shown. The entire output includes all sensor nodes and for each 4 measurements.

**$Q_3$: What is the average of the measurements for sensor** A **during 15th March 2020?**
Let $t_c$ be the timestamp defined as "2020-03-15T23:59", thus it is the end of 15th March and let $t_h$ be equal to one hour as in $Q_2$. Then $Q'_3(v,n) := \exists t(t_c - 24 \cdot t_h \leq t \leq t_c \wedge (t,v) \in \mathsf{TS}(n))$ gives us all values measured during the specific day in node $n$. Using standard notation from logic, let $Q'_3(v,n)[n = A]$ be the evaluation of this formula in the constant node $n = A$. Then, we obtain

$$Q_3(v) := \mathsf{AVG}[Q'_3(v,n)[n = A]].$$

In Cypher, this query is expressed as follows:

```
1  MATCH (n:Series {id:"A"})
2  CALL neo4j_tquery.getValuesContinuousRange(
3      n, "level",
4      left(toString(datetime("2020-03-15T23:59")
5          -duration({days:1})), 19),
6      "2020-03-15T23:59")
7  YIELD timestamp AS t, value AS v
8  RETURN avg(v);
```

As with the previous queries, the result consists of a stream of records, with in this case only one record being the average value found. For example for this query it is: 12.39, see also Figure 7.

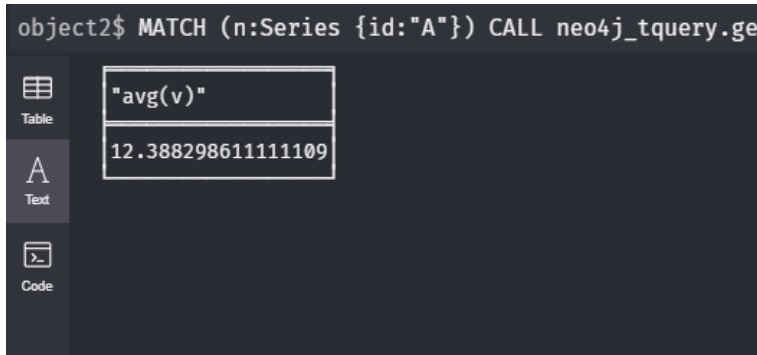

**Figure 7.** Visualization of the output of query $Q_3$ run on the sensor network of Figure 4. The result is taken from the web interface of Neo4j.

**$Q_4$: What was the highest measured value in the network during January, at which Node was that?**

In Example 12, we have already defined, assuming that January is given by the time interval $[t^-_{Jan}, t^+_{Jan}]$, the $\mathcal{TSC}$-formula $Jan(t, v, n) := t^-_{Jan} \leq t \leq t^+_{Jan} \wedge (t, v) \in \mathsf{TS}(n)$, which selects, for each sensor node $n$, the part of its time series belonging to January. Then, $Q'_4(v, n) := \mathsf{MAX}[Jan(t, v, n)]$ produces couples $(\bar{v}, n)$, where $\bar{v}$ is the maximum of the time series for January of node $n$. Then $Q_4(v, n) := Q'_4(v, n) \wedge \forall v' \forall n' (Q'_4(v', n') \to v' \leq v)$ expresses the above query $Q_4$. Obviously, it can output multiple couples, since the maximum may occur at more than one node.

In Cypher, this query is expressed as follows:

```
1  MATCH (n:Series)
2  CALL neo4j_tquery.getValuesContinuousRange(n, "level",
3      "2020-01-01T00:00", "2020-01-31T23:59")
4  YIELD timestamp AS t, value AS v
5  RETURN n.id, max(v) AS val
6  ORDER BY val DESC LIMIT 1;
```

This query effectively retrieves almost always the result wanted (see Figure 8), but it is not the perfect query. The main reason for this is the fact that selecting a maximum is performed by sorting the results on the value and then selecting the first row. This simulates maximum selection, but is not entirely correct. The method, for example, does not correctly yield the result if there are two or more locations with the same maximum value. It is clear that a more advanced maximum function is needed. However, this is a subject for future developments and implementations.

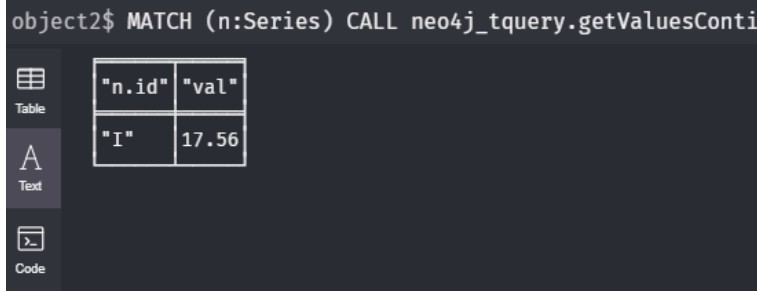

**Figure 8.** Visualization of the output of query $Q_4$ run on the sensor network of Figure 4. The result is taken from the web interface of Neo4j.

**$Q_5$: What are the last 10 values of the nearest upstream sensor for node E?**

Let us, first of all, define the predicate $NearestUpStreamSensor(n, n')$ that expresses that $n'$ is a nearest upstream sensor of node $n$. We write, using the abbreviation $\mathsf{UpStream}(n, n') := \mathsf{DownStream}(n', n)$,

$$NearestUpStreamSensor(n, n') := S(n') \wedge \mathsf{UpStream}(n, n') \wedge$$
$$\neg \exists n''(S(n'') \wedge n' \neq n'' \wedge \mathsf{UpStream}(n, n'') \wedge \mathsf{UpStream}(n'', n')).$$

We remark that we explicitly used the predicate $S$ in the above formula, as no time-series predicate is used to ensure that nodes are sensor nodes. Using the above expression, $NearestUpStreamSensor(n, n')[n = \mathsf{E}]$ defines $n'$ as a nearest upstream sensor of node $E$.

The last 10 values of a sensor node $n$ can then be obtained (by a long formula) using the predicates $LastTime(t, n)$ and $NextTime(t, t', n)$ of Examples 5 and 6 (or variants of these that use values instead of time). We omit the details.

In Cypher, this query is expressed as follows:

```
1  MATCH path = (n:Segment {id:"E"})<-[r:to*]-(m:Series)
2  WITH  m, length(path) AS l
3  ORDER BY l LIMIT 1
4  CALL neo4j_tquery.getValuesDiscreteRange(m, "level",
5      0, 10)
6  YIELD timestamp AS t, value AS v
7  RETURN t, v;
```

The result of the query can be seen in Figure 9. In this query, the path length is used in order to find the nearest sensor. This is in line with the logic given earlier. Because the shortest path means that there is no other sensor between this one and the start of the path. We notice that, with this implementation for selecting the shortest path, the same observation as at the end of query $Q_4$ can be made.

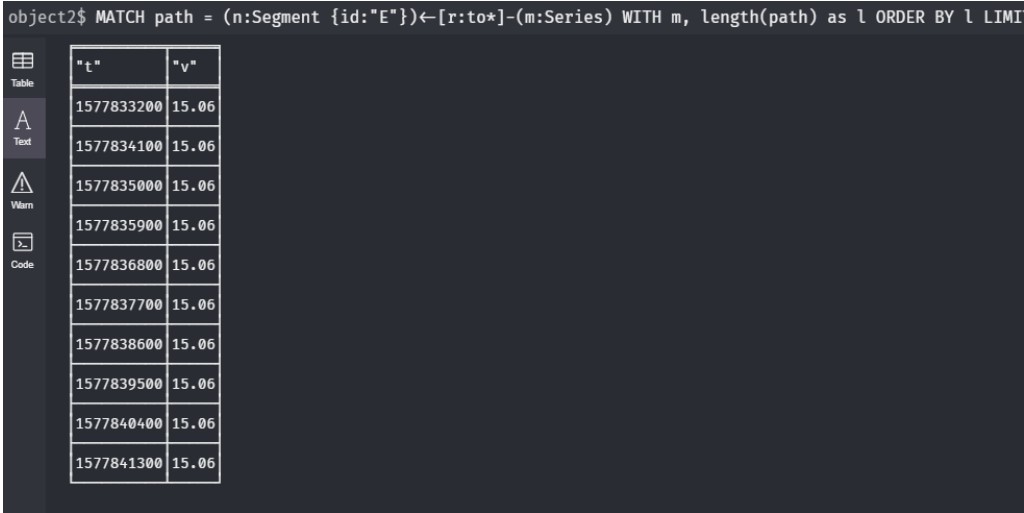

**Figure 9.** Visualization of the output of query $Q_5$ run on the sensor network of Figure 4. The result is taken from the web interface of Neo4j.

**$Q_6$: What are the upstream sensor values at the moment that location G has its maximal value during 15th March?**

We assume that 15th March is given by the time interval $[t^-_{Mar15}, t^+_{Mar15}]$. The maximal value of a sensor during that day is given by the expression $MaxV_{Mar15}(v, n) := \mathsf{MAX}(t^-_{Mar15} \leq t \leq t^+_{Mar15} \wedge (t, v) \in \mathsf{TS}(n))$. The time $t$ this maximum could be reached is then given by $MaxT_{Mar15}(t, n) := \exists v((t, v) \in \mathsf{TS}(n) \wedge MaxV_{Mar15}(v, n))$. Thus, the time this maximum is reached during 15th March in node G is given by $MaxT@G_{Mar15}(t) := MaxT_{Mar15}(t, n)$ $[n = \mathsf{G}]$. Thus, query $Q_6$ is expressed as

$$Q_6(v, n) := \mathsf{UpStream}(\mathsf{G}, n) \wedge S(n) \wedge \exists t (MaxT@G_{Mar15}(t) \wedge (t, v) \in \mathsf{TS}(n)).$$

In Cypher, this query is expressed as follows:

```
1  MATCH (n:Series {id:"G"})
2  CALL neo4j_tquery.getValuesContinuousRange(n, "level",
3     "2020-03-15T00:00", "2020-03-15T23:59")
4  YIELD timestamp AS t, value AS v
5  WITH n, t AS tg, v AS vg ORDER BY vg DESC LIMIT 1
6  MATCH (n)<-[:to*]-(m:Series)
7  RETURN m.id, neo4j_tquery.getValueContinuous(m, "level",
8     left(toString(datetime({epochSeconds: tg})), 19));
```

This cypher query first matches the node with ID *G* and then determines the timestamp, *tg*, at which the value reaches its highest value during the specified day. With this information, the second part matches all sensors that are upstream of the node *G* and then retrieves, for all those sensors, the value they have at moment *tg*. In this case, the results are $(I, 18.41), (J, 3.63), (N, 0.816)$, as can be seen in Figure 10. Finally, the remarks have to be made that implemented procedures return timestamps as epoch timestamps, therefore there is some conversion when *tg* is supplied as parameter and the maximum selection in part one suffers the same problems discussed in $Q_4$ and $Q_5$.

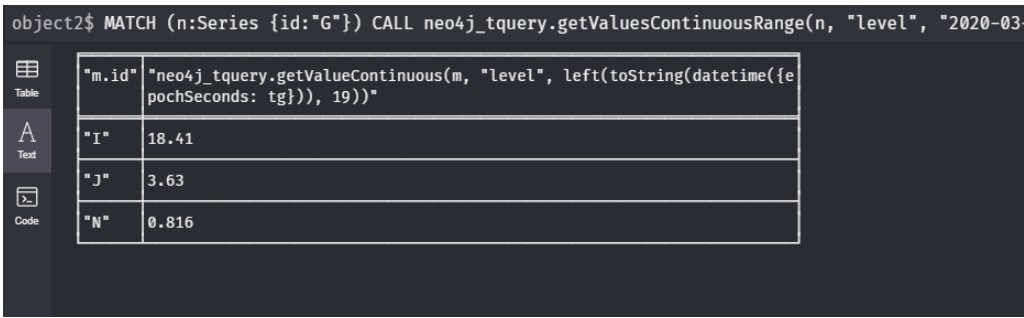

**Figure 10.** Visualization of the output of query $Q_6$ run on the sensor network of Figure 4. The result is taken from the web interface of Neo4j.

**$Q_7$: What is the average value in each series for all downstream sensors of a location F?**
The formula $S(n) \land \text{DownStream}(n', n))[n' = \text{F}]$ defines the set of sensor nodes *n* that are downstream with respect to the node F. With this in mind, all values for those nodes need to be selected. Therefore, a separate predicate is defined and then the average can be taken.

$$VDownF(t, v, n) := (t, v) \in \text{TS}(n) \land S(n) \land \text{DownStream}(n', n)[n' = \text{F}]$$

$$Q_7 := AVG[VDownF(t, v, n)]$$

In Cypher, this query is expressed as follows:

```
1  MATCH (n:Segment {id:"F"})-[:to*]->(m:Series)
2  CALL neo4j_tquery.getValuesContinuousRange(
3     m, "level", "epoch", "now")
4  YIELD timestamp AS t, value AS v
5  RETURN m.id, avg(v);
```

This query uses different aspects introduced earlier such as aggregation and downstream or upstream matching. However, in addition, it demonstrates the keywords "now" and "epoch" which stand for the current timestamp and "1970-01-01T00:00:00", respectively. The epoch timestamp is used in this work as the beginning of time. The result of this query consists of three records, namely, $(A, 9.95), (B, 12.03), (D, 11.95)$, see Figure 11.

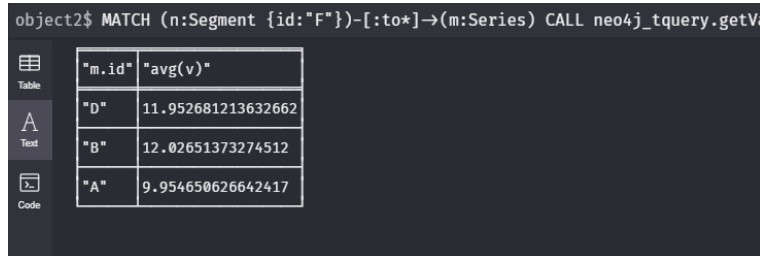

**Figure 11.** Visualization of the output of query $Q_7$ run on the sensor network of Figure 4. The result is taken from the web interface of Neo4j.

**$Q_8$: What is the time difference between the moments where sensor G and D reach their maximum during the last day of 2020?**

As with $Q_6$, we assume that the indicated day is given by the time interval $[t_{Dec31}^-, t_{Dec31}^+]$ and we can use an expression $MaxT_{Dec31}(t, n)$, similar to the one we made for $Q_6$. Then, $MaxT@G_{Dec31}(t) := MaxT_{Dec31}(t, n)[n = \text{G}]$. and $MaxT@D_{Dec31}(t) := MaxT_{Dec31}(t, n)[n = \text{D}]$ give the two moments.

Thus, query $Q_8$ is expressed as

$$Q_8(t) := \exists t' \exists t'' (MaxT@G_{Dec31}(t') \land MaxT@D_{Dec31}(t'') \land$$
$$((t' - t'' \geq 0 \land t = t' - t'') \lor (t' - t'' \leq 0 \land t = t'' - t'))).$$

In Cypher, this query is expressed as follows:

```
1  MATCH (n:Series {id:"G"})
2  CALL neo4j_tquery.getValuesContinuousRange(n, "level",
3      "2020-12-31T00:00", "2020-12-31T23:59")
4  YIELD timestamp AS t, value AS v
5  WITH t AS tg, v AS vg ORDER BY vg DESC LIMIT 1
6  MATCH (m:Series {id:"D"})
7  CALL neo4j_tquery.getValuesContinuousRange(m, "level",
8      "2020-12-31T00:00", "2020-12-31T23:59")
9  YIELD timestamp AS t, value AS v
10 WITH tg, vg, v AS vd, t AS td ORDER BY vd DESC LIMIT 1
11 RETURN duration.between(
12     datetime({epochSeconds: td}), datetime({epochSeconds: tg}));
```

Because the procedures return (timestamp, value) pairs, it is possible to operate on the timestamps themselves. Again, selecting the maximum value here is not the final solution envisioned, but the query clearly illustrates that asking temporal questions is possible. The time and duration type and functions of Neo4j already provide almost all the functionality needed to calculate the difference between the timestamps. First, the two nodes are matched and the timestamp during which the maximum is occurring is selected. Again, the maximum selection procedure is not valid for all cases. Finally, the difference between those two timestamps is returned, which is in this case: 9900 s or 2 h and 45 min. This is shown in Figure 12.

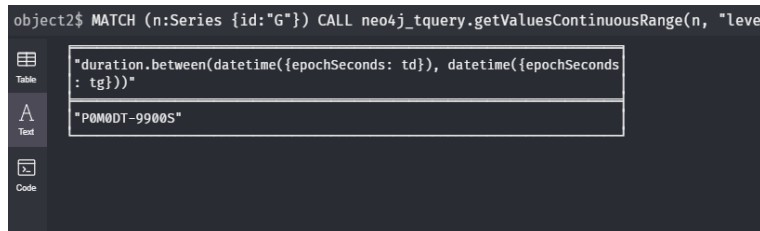

**Figure 12.** Visualization of the output of query $Q_8$ run on the sensor network of Figure 4. The result is taken from the web interface of Neo4j.

**Q₉: Do all sensors on a path from J to A have all measurements below a threshold value $\tau$ during interval I?**

We write this query by an expression that depends on the parameter $\tau$. Let $I = [t^-, t^+]$ represent the time interval I. Then, the query $Q_9$ is expressed as follows:

$$Q_9(\tau) := \forall n((S(n) \wedge \mathsf{DownStream}(\mathsf{J}, n) \wedge \mathsf{DownStream}(n, \mathsf{A})) \rightarrow$$
$$\forall t \forall v(t^- \leq t \leq t^+ \wedge (t, v) \in \mathsf{TS}(n) \rightarrow t \leq \tau).$$

In Cypher, this query is expressed as follows, taking the month March as the time range and 20 as the threshold:

```
1  MATCH path =
2  (n:Segment {id:"J"})-[:to*]->(s:Series)-[:to*1..12]->(m:Segment {id:"A"})
3  CALL neo4j_tquery.getValuesContinuousRange(
4      s, "level", "2020-03-01T00:00", "2020-03-31T01:00")
5  YIELD timestamp AS t, value AS v
6  WITH path AS p, collect(v) AS vs
7  WHERE all(i IN vs WHERE i < 20)
8  RETURN count(p) >= 1;
```

The question originally asks if there is a path where all sensors stay below a threshold. The Cypher will find a path adhering to this condition if there is one. Therefore, if the result contains one or more paths, the answer is true and false otherwise, see Figure 13. The result of the query for this data evaluates to true and would evaluate to false for a threshold that is 18 or smaller.

We note that the recursive relationship matching is limited between 1 and 12 relationships. This is not required by the query, but this query does not work on use cases where the path is longer than 12 hops. However, Neo4j incurs large performance reductions when unlimited recursion matching is used. In order to reduce the possible run time, this extra condition was added.

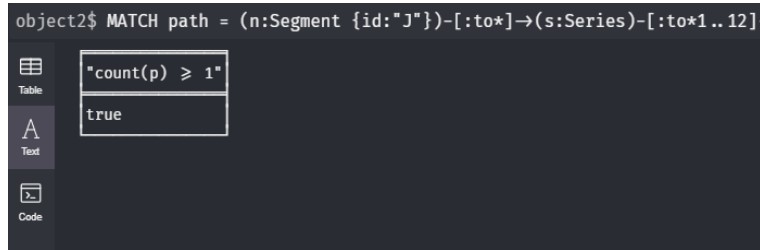

**Figure 13.** Visualisation of the output of query $Q_9$ run on the sensor network of Figure 4. The result is taken from the web interface of Neo4j.

**Q₁₀: Return the paths where the sensors on a path from J to A have all measurements below a threshold value $\tau$ during interval I?**

This is the same query as $Q_9$, but instead of returning a Boolean value, this time the (nodes belonging to the) paths are the result of the query. It illustrates that path selection with a time-series condition on the path is possible. This illustrates the idea of queries that combine the two dimensions into one. It is possible to split the queries in a temporal and spatial component and run them separately. However, the goal of this idea is to take away from the user this burden and supply him/her with the ability to just ask for the combination. Later on, this should also give the possibility to create optimizations that are in the two separate cases not possible because the conjunction of the condition can eliminate certain cases.

The query can be expressed in $\mathcal{TSC}$ as follows: $Q_{10}(\tau, n) := \mathsf{DownStream}(\mathsf{J}, n) \wedge \mathsf{DownStream}(n, \mathsf{A}) \wedge Q_9(\tau)$. We note that, this formula returns a set of nodes, which can form (multiple) paths between the indicated nodes J and A.

Expressing this in Cypher, with the same parameter as in $Q_9$, yields

```
1  MATCH path =
2  (n:Segment {id:"J"})-[:to*]->(s:Series)-[:to*1..12]->(m:Segment {id:"A"})
3  CALL neo4j_tquery.getValuesContinuousRange(
4      s, "level", "2020-03-01T00:00", "2020-03-31T01:00")
5  YIELD timestamp AS t, value AS v
6  WITH path AS p, collect(v) AS vs
7  WHERE all(i IN vs WHERE i < 20)
8  RETURN p;
```

The result consists of all paths $p$ that are adhering to the conditions. In Neo4j, a path is expressed as a set of Nodes and the edges between them. The real output is shown in Figure 14. This is slightly different than the output defined in the $\mathcal{TSC}$ formula, but in essence it can be considered the same with some conversions. This shows that there is still some work left to coincide the theory with the practice in up-coming work.

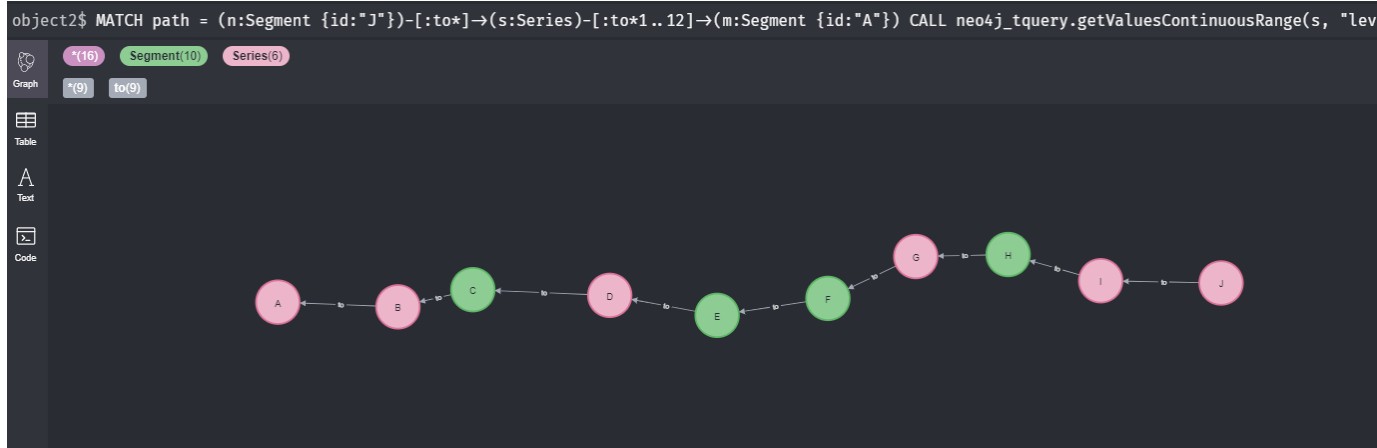

**Figure 14.** Visualization of the output of query $Q_{10}$ run on the network, visualized in Figure 4. The result is taken from the web interface Neo4j. All nodes correspond to river segments, and the segments equipped with a sensor are indicated in pink. In this case, there is only one path but the query would return multiple if they exist.

## 6. Discussion and Future Work

In this paper, we consider the idea of *querying and managing data related to a sensor-equipped stable transportation network*. Such a network is represented by a graph, which reflects the spatial dimension, and time series linked to the nodes of the graph, representing the temporal dimension of the data. On this data model, a logic-basic formalism is introduced where, in addition to the basic first-order logic, recursion and aggregation are added in order to express domain-relevant queries. The *time-series calculus*, $\mathcal{TSC}$, is then used together with free time-variables, value-variables, and node-variables to match patterns that adhere to the conditions expressed. The patterns that are matched are then considered the result of the expression, or the query, in general. This idea is further demonstrated through a proof-of-concept implementation over the Neo4j graph database. In this implementation, the query language Cypher is extended with user-defined functions and procedures. Over this implementation, a use case studies the river system in Flanders using water-level data to pose hydrological questions. Those questions are translated into $\mathcal{TSC}$ and Cypher before they are run and discussed. Altogether, these parts are a proposal to fill the gap between spatiotemporal data and the queries to them. Thus, we build the foundation for more elaborate and detailed work of the theory or practice and our work is the basis for more advanced analytical developments.

The transportation network used in this paper is simple: the nodes of the graph are equipped with at most one sensor, measuring one numeric value. In the future, we plan to extend this model to associate both nodes and edges with any number of sensors,

measuring any number of values of any type. This means a node could have multiple time series attached to it, but also a time series with multiple values per time point, or even series with entire objects as values (for example, geometries). In addition, allowing multiple time series to nodes or edges would be more representative of real-world situations, as sensors often measure multiple parameters at the same time (e.g., for river systems sensors can provide time series for salinity, pollution, among other ones). Furthermore, we want to extend the model to time series of any origin, not only restricted to sensors, e.g., opening hours of shops, time tables of public transport, etc. All of the above also impacts on the choice of possible representation for the implementation. Three possible approaches were mentioned in Section 5.2, namely, the object, linked-list, and external TSMS approaches. Although the object approach is used in this paper, it remains to be confirmed if this is the most appropriate one. The additional developments, described here, also influence this research. For example, if series need to be linked to relations (and not only nodes), it would be more difficult to use the linked-list approach, because it is impossible to link a node (the linked list) to a relation, using an additional relation. Choosing an external TSMS also implies that the functionality and query capabilities of this system must be taken into account. Deciding these questions will be also addressed in future work.

The implementation of the proposed formalism in the Cypher language is a proof of concept of how a language can be realized over a data management system. However, the version shown here has some limitations from a real-world user's point of view. For example, although the queries can be completed by the Neo4j query engine within milliseconds, the results are not displayed that fast. This is because the properties of the nodes (that is, entire series) are held in memory by the Neo4j interface, which slows down the rendering of the result. Furthermore, the functionality available for series in the system is limited to what was needed for this work. For instance, it is possible to retrieve data but not to apply advanced functions to them (for example, taking a moving average). However, this can be solved by extending the systems with the desired functions similar to the PostGis extension or by combining Cypher with another query language, for example, Flux. A more general approach would be to define a new query language that can handle graph and time-series data together. In this sense, a more general problem would be to define whether we need a graph query language extended with a temporal dimension, or a temporal language extended to support graph data. From a theoretical point of view, we remark that some specific queries cannot be expressed in the presented calculus. Examples are queries that count the number of nodes or segments within a path. This and other constructions, for example the assignment operator, will be addressed in future work.

**Author Contributions:** Erik Bollen, Rik Hendrix, Bart Kuijpers and Alejandro Vaisman contributed equally to all aspects of this paper. All authors have read and agreed to the published version of the manuscript.

**Funding:** The research of Erik Bollen was supported by the *Bijzonder Onderzoeksfonds* (BOF) from UHasselt with reference BOF20OWB27 and by VITO with project reference 2010478. Alejandro Vaisman was partially supported by Project PICT 2017-1054, from the Argentinian Scientific Agency.

**Institutional Review Board Statement:** Not applicable.

**Informed Consent Statement:** Not applicable.

**Data Availability Statement:** The data presented in this study are openly available at OSF with DOI 10.17605/OSF.IO/95TCX.

**Conflicts of Interest:** The authors declare no conflicts of interest.

**Appendix A**

This safety query $\sigma_\varphi$ is based on the following lemma.

**Lemma A1.** *Let $\mathcal{SN} = (S, N, E, \mathsf{TS})$ be a sensor network and let $N_0 \subseteq N$ be a set of nodes that satisfies the following conditions:*

1.  $N_0$ has exactly one node (call it $n_b$) with indegree 0 and outdegree 1 in $N_0$ (With "in $N_0$", we mean the in- and outdegree in the subgraph of $(N, E)$ induced by $N_0$.);
2.  $N_0$ has exactly one node (call it $n_e$) with indegree 1 and outdegree 0 in $N_0$;
3.  all nodes in $N_0 \setminus \{n_b, n_e\}$ have indegree 1 and outdegree 1 in $N_0$.

If Conditions $1 - 3$ are satisfied, then there is a path in $N$ starting from $n_b$ and ending in $n_e$ that visits all the nodes of $N_0$.

**Proof.** From the conditions, it follows that $N_0$ has at least two nodes, namely, $n_b$ and $n_e$. We prove this lemma by induction on the cardinality of $N_0$. For the case, $|N_0| = 2$, we have $N_0 = \{n_b, n_e\}$ and as $n_b$ has outdegree 1 in $N_0$, there must be an edge between $n_b$ and $n_e$. This edge is a path that visits all nodes in $N_0$. Suppose the lemma holds for $|N_0| = k$. We need to show that it holds for $|N_0| = k + 1$. Let $N_0$ be a set of nodes that satisfies the conditions of the lemma. As $N_0$ has exactly one node $n_b$ with indegree 0 and outdegree 1, there must be a node $n'_b$ in $N_0$ such that the outgoing edge from $n_b$ arrives there. Now, we consider $N'_0 = N_0 \setminus \{n_b\}$, which has cardinality $k$. As in $N_0$ there was an edge between $n_b$ and $n'_b$, removing $n_b$ from $N_0$ results in $n'_b$ having indegree 0 and outdegree 1 (in $N'_0$). For the remaining nodes the conditions remain true and we can apply the induction hypothesis which gives us a path from $n'_b$ to $n_e$ visiting all nodes in $N'_0$. When we add the edge between $n_b$ and $n'_b$ we obtain a path from $n_b$ to $n_e$ visiting all nodes in $N_0$. This finishes the proof. $\square$

With this lemma in mind, we can start to construct the safety formula $\sigma_\varphi$.

The following $\mathcal{TSC}$-formula $out^1_\varphi(n)$ expresses that the outdegree in $N_\varphi$ of a node $n$ (that belongs to $N_\varphi$) is 1:

$$out^1_\varphi(n) := \varphi(n) \wedge \exists n'(\varphi(n') \wedge E(n, n') \wedge \forall n''((\varphi(n'') \wedge E(n, n'')) \to n' = n'')).$$

The following formula $out^0_\varphi(n)$ expresses that the outdegree in $N_\varphi$ of a node $n$ (that belongs to $N_\varphi$) is 0:

$$out^0_\varphi(n) := \varphi(n) \wedge \neg \exists n'(\varphi(n') \wedge E(n, n')).$$

Similarly, we can construct formulas $in^1_\varphi(n)$ and $in^0_\varphi(n)$ that express that the indegree in $N_\varphi$ of a node $n$ (that belongs to $N_\varphi$) is 1, respectively 0:

$$in^1_\varphi(n) := \varphi(n) \wedge \exists n'(\varphi(n') \wedge E(n', n) \wedge \forall n''((\varphi(n'') \wedge E(n'', n)) \to n' = n''))$$

and

$$in^0_\varphi(n) := \varphi(n) \wedge \neg \exists n'(\varphi(n') \wedge E(n', n)).$$

Keeping the conditions of Lemma A1 in mind, the safety formula $\sigma_\varphi$ can then be written as

$$\sigma_\varphi := \exists n_b \exists n_e \Big( out^1_\varphi(n_b) \wedge in^0_\varphi(n_b) \wedge out^0_\varphi(n_e) \wedge in^1_\varphi(n_e) \wedge$$
$$\forall n((\varphi(n) \wedge n \neq n_b \wedge n \neq n_e) \to (out^1_\varphi(n) \wedge in^1_\varphi(n)))\Big).$$

The safety formula $\sigma_\varphi$ evaluates to true if and only if the set of nodes $N_\varphi$, defined by $\varphi(n)$, defines a path in the sensor network.

This leads to the Proposition 2: If the formula $\varphi(n)$ expresses a unary node selection query, then the formula

$$\varphi(n) \wedge \sigma_\varphi$$

expresses a path selection query.

We remark that when $\varphi(n)$ does not define a path, the formula $\varphi(n) \wedge \sigma_\varphi$ will return the empty path (empty set of nodes). We also remark that any path selection query, that is based on a unary node selection query, can be written in the form $\varphi(n) \wedge \sigma_\varphi$.

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
