# Peer review of "Time-Series-Based Queries on Stable Transportation Networks Equipped with Sensors"

_ijgi, doi:10.3390/ijgi10080531_

Round 1

Reviewer 1 Report

Dear Authors,

thanks for the nice paper that addresses the issue of adding to monitoring networks the logic of connectivity. The issue is approached from a conceptual point of view and mathematically formalized (TSC). Then it is implemented in a proof-of-concept system based on neo.js and applied in a case study (Flander river system) to evaluate results.​ ​

The paper is not of easy reading due to the large number of formalis​​ms introduced, but is accurate and I guess mostly required. Nevertheless, the authors could maybe consider if some of them could be moved in appendixes to facilitate the readings.

While in the introduction the option of handling the network in the graph DB and the time series in the time series management systems is mentioned, it wasn't the selected choice. I wonder if you could elaborate the rationale of your choice since most of the implemented TS features (aggregation and filtering on time and values) are available in TSMS or more easily implemented. It could be also appropriate to discuss this in the final conclusions.

In your work, you considered only basic TS of key, value pairs composed by numeric values. Have you thought about how you could extend your concept from 1D observation (like river heights) to 2D observations (like velocity profiles or images)​or more dimensions?​? This could also be shortly addressed / mentioned in the final discussions.

You also mentioned geospatial queries, but you didn't elaborate on this topic. Will a logic graph be enough to comply with transportation management issues? How the lack of edges and vertex location could affect the exploitation of such a system? (length of edges, intersections with other systems, etc...).

At page 16 you opted to return the last observed value for a requested time instant. This may be a questionable choice, since its approximation with the requested value in time depends on several aspects like data completeness, observed phenomenon and sampling resolution. Could you elaborate it more on how you consider these aspects (missing data, resolution, observed properties..)?

Since the journal you submitted to is a geoscience journal should keep this dimension in consideration. At minimum I would consider geoscience related works in the Related Work sections and would extend more the discussions on this topic.

Some of the papers you should for example consider are:

- Ventura, B., Vianello, A., Frisinghelli, D., Rossi, M., Monsorno, R., & Costa, A. (2019). A Methodology for Heterogeneous Sensor Data Organization and Near Real-Time Data Sharing by Adopting OGC SWE Standards. ISPRS International Journal of Geo-Information8(4), 167.

- Cannata, M., Antonovic, M., Molinari, M., & Pozzoni, M. (2015). istSOS, a new sensor observation management system: software architecture and a real-case application for flood protection. Geomatics, natural hazards and risk6(8), 635-650.

- Liang, S., Huang, C. Y., & Khalafbeigi, T. (2016). OGC SensorThings API Part 1: Sensing, Version 1.0.

Author Response

Comment 1:

The paper is not of easy reading due to the large number of formalisms introduced, 

but is accurate and I guess mostly required. Nevertheless, the authors could maybe consider if 

some of them could be moved in appendixes to facilitate the readings.

 Reply 1:We understand the reviewer's concerns, although we would like to mention that query languages 

 for Constraint Databases are typically formulated in terms of first-order logic over real numbers. Therefore, 

 papers on the topic usually have this theoretical flavour (see, e.g., Kuper,   Libkin, Paredaens: Constraint Databases. Springer, 2000. Since  

 this paper was submitted to an IJGI special issue on ``Spatio-Temporal and Constraint Databases'', we think the audience  will 

 expect these logic-based formalisms. Anyway, we tried to keep  the formalism to the minimum possible to make the paper 

 self-contained. We also tried to make it possible to follow the main ideas conveyed by the paper skipping the formal part 

 (which, basically, spans Section 3). Therefore, we respectfully consider that for completeness and consistency reasons, 

 we would prefer to keep the formal part as it is now, since   there are no large parts that can be moved away from the main text.

Comment 2: While in the introduction the option of handling the network in the graph DB and the time series in 

the time series management systems is mentioned, it wasn't the selected choice. I wonder if you could elaborate 

the rationale of your choice since most of the implemented TS features (aggregation and filtering on time and values) 

are available in TSMS or more easily implemented. It could be also appropriate to discuss this in the final conclusions. 

 Reply 2:Thank you for the suggestion. We added an extensive additional explanation in Sections 5.2 

 (Data model and Cypher extensions) and Section 6 (Discussion and future work) to clarify our   selection of the object approach.

Comment 3:  In your work, you considered only basic TS of key, value pairs composed by numeric values. Have you thought 

about how you could extend your concept from 1D observation (like river heights) to 2D observations (like velocity profiles or images) 

or more dimensions?  This could also be shortly addressed / mentioned in the final discussions.

 Reply 3:Thank you for this remark. Indeed, extending our basic approach  is planned as future work, in which we will also allow not 

 only key-values as time series, but   general key-objects (so, the time series could be composed of any kind of object). 

 We have added a comment, clarifying this issue  in Section 6 (Discussion and future work).

Comment 4:   You also mentioned geospatial queries, but you didn't elaborate on this topic. Will a logic graph be enough 

to comply with transportation management issues? How the lack of edges and vertex location could affect the exploitation of 

such a system? (length of edges, intersections with other systems, etc...).

 Reply 4: Actually we mean that queries could mention places of interest or geographic places  as we show in [34]. 

 In Sections    5.2 and 6,  we briefly commented on the possibility of adding geo-locations of network segments. 

Comment 5: At page 16 you opted to return the last observed value for a requested time instant. 

This may be a questionable choice, since its approximation with the requested value in time depends on 

several aspects like data completeness, observed phenomenon and sampling resolution. 

Could you elaborate it more on how you consider these aspects (missing data, resolution, observed properties..)?

Reply 5: Thank you for this  very relevant observation. The aspects mentioned need to be taken into consideration 

when interpolation or approximation functionalities are performed. However, these aspects also depend on the 

application or use case at hand. The goal is to develop a generic system, and therefore, the choice regarding the 

exact interpolation or approximation is reserved to the user. The complete implementation is foreseen as future work. 

In this paper, we chose to add a simple interpolation scheme for the ease of use, although it would also be possible to 

eliminate any interpolation functionality for the use case presented here. We also want to remark that this functionality 

is commented on in Section 5.2 (Data model and Cypher extensions),  although the queries  included  in this paper make no use of it.

Comment 6:    Since the journal you submitted to is a geoscience journal should keep this dimension in consideration. 

At minimum I would consider geoscience related works in the Related Work sections and would extend more the discussions on this topic.

Some of the papers you should for example consider are: 

- Ventura, B., Vianello, A., Frisinghelli, D., Rossi, M., Monsorno, R., & Costa, A. (2019). A Methodology for Heterogeneous 

Sensor Data Organization and Near Real-Time Data Sharing by Adopting OGC SWE Standards.

ISPRS International Journal of Geo-Information, 8(4), 167. 

- Cannata, M., Antonovic, M., Molinari, M., & Pozzoni, M. (2015). istSOS, a new sensor observation management system: 

software architecture and a real-case application for flood protection. Geomatics, natural hazards and risk, 6(8), 635-650. 

- Liang, S., Huang, C. Y., & Khalafbeigi, T. (2016). OGC SensorThings API Part 1: Sensing, Version 1.0.

Reply 6: Thank you for these additional resources. We studied them and included them in Section 2 (Related Work). 

We also included a reference mentioned in the two papers you suggested.

Reviewer 2 Report

Authors present really important topic regarding the data management. The contribution of the paper is clear and structure looks good, as well. 

I have some minor comments to be applied: 

Regarding the analysis, I can point your attention to the following paper dealing with the various temporal architectures: https://www.mdpi.com/2076-3417/11/3/916/htm. I recommend to discuss such paper and reference it.

Other references relevant to the paper are:

Wang, S., Zhang, Z., Wu, Z., Liu, J., Mo, C.: Driver Drowsiness Analysis Based on Eyelid Feature. In: Lecture Notes in Electrical Engineering, 2021 

Ehlers, U.C., Ryeng, E.O., McCormack, E., Khan, F., Ehlers, S.: Assessing the safety effects of cooperative intelligent transport systems: A bowtie analysis approach. In: Accident Analysis and Prevention, 2017

Sanchez, C.S., Wieder, A., Sottovia, P., Bortoli, S., Baumbach, J., Axenie C.: GANNSTER: Graph-Augmented Neural Network Spatio-Temporal Reasoner for Traffic Forecasting, Advanced Analytics and Learning on Temporal Data, 5th ECML PKDD Workshop, AALTD 2020 Ghent, Belgium, September 18, 2020

Peng, T., Sellami, S., Boucelma, O.: Trust Assessment on Streaming Data: A Real Time Predictive Approach, Advanced Analytics and Learning on Temporal Data, 5th ECML PKDD Workshop, AALTD 2020 Ghent, Belgium, September 18, 2020

Pornphol, P., Chittayasothorn, S.: Temporal database application in industrial human resource management. In: CIE 2014 - 44th International Conference on Computers and Industrial Engineering, 2014

Related work and introduction sections are complex, maybe, they can be shortened pointing just to the main contribution of the paper. 

Main limitation of the paper is regarded to the figures, I recommend to comment them in textual form more properly, mostly fig. 1, fig. 2 and fig. 3. It is not clear, what the letter inside the fig. 3 mean. 

Code in line 576 does not fit the paper size. 

Output visualization figures are hard to be read. Maybe, rewrite them to the table form with better readability option. 

Extend the discussion section to be more specific to the results and limitations. In line 764, you mention the limitations. Please be more specific and summarize them in textual form. 

Add the paragraph dealing with the real usage of the proposed techniques - some example case study. 

There are some small points to improve the paper to become excellent. 

Thank you for your contribution. 

Author Response

 Comment 1: Regarding the analysis, I can point your attention to the following paper dealing with the various temporal architectures: 

 https://www.mdpi.com/2076-3417/11/3/916/htm. I recommend to discuss such paper and reference it. 

 Other references relevant to the paper are:

Wang, S., Zhang, Z., Wu, Z., Liu, J., Mo, C.: Driver Drowsiness Analysis Based on Eyelid Feature. In: Lecture Notes in Electrical Engineering, 2021

Ehlers, U.C., Ryeng, E.O., McCormack, E., Khan, F., Ehlers, S.: Assessing the safety effects of cooperative intelligent 

transport systems: A bowtie analysis approach. In: Accident Analysis and Prevention, 2017

Sanchez, C.S., Wieder, A., Sottovia, P., Bortoli, S., Baumbach, J., Axenie C.: GANNSTER: 

Graph-Augmented Neural Network Spatio-Temporal Reasoner for Traffic Forecasting, Advanced Analytics and 

Learning on Temporal Data, 5th ECML PKDD Workshop, AALTD 2020 Ghent, Belgium, September 18, 2020

Peng, T., Sellami, S., Boucelma, O.: Trust Assessment on Streaming Data: A Real Time Predictive Approach, 

Advanced Analytics and Learning on Temporal Data, 5th ECML PKDD Workshop, AALTD 2020 Ghent, Belgium, September 18, 2020

Pornphol, P., Chittayasothorn, S.: Temporal database application in industrial human resource management. 

In: CIE 2014 - 44th International Conference on Computers and Industrial Engineering, 2014

 Reply 1: Thank you for this extensive list of extra resources. After carefully checking at them, 

 we included the most relevant three for our work, in Section 2 (Related Work). 

Comment 2:  Related work and introduction sections are complex, maybe, they can be shortened pointing 

just to the main contribution of the paper.

Reply 2:The main contributions of the paper are highlighted under the paragraphs following the title Contributions 

in the Introduction. We followed the traditional approach in scientific writing, splitting the introduction and motivation, 

and the related work parts, into different sections.  Section 2 was also expanded to address the suggestions of the three reviewers.

Comment 3:   Main limitation of the paper is regarded to the figures, I recommend to comment them in textual form more properly, 

mostly fig. 1, fig. 2 and fig. 3. It is not clear, what the letter inside the fig. 3 mean.

 Reply 3:Thank you for this observation. We have extended the captions of Figures 1 and 2 to clarify what is shown there. 

 Figure 3 has been replaced by a more clear one (better background and segments highlighted). We also extended the caption in this figure.

Comment 4:     Code in line 576 does not fit the paper size.

Reply 4:Thank you. We fixed it. 

Comment 5:  Output visualization figures are hard to be read. Maybe, rewrite them to the table form with better readability option.

Reply 5:We increased the size of the figures showing the output of queries 1, 2 and 5 and 6. 

We prefer to stick to this format of screenshots, since it shows what a user can expect to see. 

For clarity, most of the outputs are commented and explained in the text.

Comment 6:  Extend the discussion section to be more specific to the results and limitations. 

In line 764, you mention the limitations. Please be more specific and summarize them in textual form.

 Reply 6:Thank you for the observation. We added a discussion on the implementation and limitation of 

 the representation used, as well as examples of the limitation of the proposed theory in Section 6 

 (Discussion and future work) and partially in Section  5.2 (Data model and Cypher extensions). 

 With respect to practical results of the queries executed, we have added a small discussion and an 

 idea of the execution times (as also requested by Reviewer 3), but we have not further elaborated on this topic, 

 because we wanted to focus on the model and proposed language, and not on the Neo4j implementation. 

 Addressing execution times is not the main point of this paper.

Comment 7:  Add the paragraph dealing with the real usage of the proposed techniques - some example case study.

 Reply 7:We showed the feasibility of our proposal with a small use case based on the Flanders river system presented in   

 Section 5. This is a real-world case study in the framework of the "Internet of Water" project. We included an additional 

 clarification for this at the beginning of Section 5.    The river use case is a first example of how this idea 

 can be applied in practice.  Adding other use cases would make the paper unnecessary longer.

Reviewer 3 Report

Several recent references approach the subject of this paper: the authors should refer them to substantiate their thesis. In the following I suggest several references that are related, it is up to the authors to check which are the most useful to better clarify the context of the paper and support their thesis.

 When the authors say in rows 13-16: “There are many applications for sensor networks, from monitoring a single home, to the surveillance of a large city, to earthquake detection for the whole world. [ref1] For example, researchers, farmers, and governments need to monitor aspects of the natural environment such as air pollution, water quality, soil conditions, and weather metrics” [ref2]

Several recent references this issue. Among others:

  • [ref1]
  • Cozza V., Guagliardi I., Rubino M., Cozza R., Martello A., Picelli M., Zhupa E., 2015. Esopo: Sensors and social pollution measurements. CEUR Workshop Proceedings, 1478, 52-57.
  • Ma, Yajie, Guo, Yike, Ghanem, Moustafa: Distributed pattern recognition for air quality analisys in sensor network system. IADIS International Journal on Computer Science and Information Systems Vol. 5, No.1, pp. 87-100ISSN:1646-3692
  • [ref2]
  • Buttafuoco G., Guagliardi I., Tarvainen T., Jarva J., 2017. A multivariate approach to study the geochemistry of urban topsoil in the city of Tampere, Finland. Journal of Geochemical Exploration 181, 191-204
  • - Zuzolo D., Cicchella D., Lima A., Guagliardi I., Cerino P., Pizzolante A., Thiombane M., De Vivo B., Albanese S., 2020. Potentially toxic elements in soils of Campania region (Southern Italy): Combining raw and compositional data. Journal of Geochemical Exploration, 213, Article number 106524

The authors say: “The core of the proposed query mechanismis a logic-based3 language that is capable to return time, value and time series outputs, as well as Boolean queries”. This issue has been investigated in several papers that take into account spatial-temporal queries, in line with the Jensen work already mentioned by the authors as [11], there are also:

  • Cozza, V., Messina, A., Montesi, D., Arietta, L., Magnani, M.: Spatio-temporal keyword queries in Social Networks. In Springer, editor, B. Catania, G. Guerrini, and J. Pokorn(Eds.): ADBIS 2013, volume 8133 of LNCS, pages 70-83, 2013.
  • Anand, A., Bedathur, S., Berberich, K., Schenkel, R.: Efficient Temporal Keyword Queries over Versioned Text. In: Proc. of ACM CIKM Conf. (2010)
  • Berberich, K., Bedathur, S., Neumann, T., Weikum, G.: A time machine for text search. In: Proceedings of the 30th Annual International ACM SIGIR Conference on Research and Development in Information Retrieval - SIGIR 2007, p. 519 (2007)

Moreover, It seems the submitted paper is a follow up work of [32], however [32] is not yet published so I cannot read on it. What is needed is that the authors at least mention how the present paper differs from that one.

This idea presented in the paper is very interesting, and the proof-of-concept implementation over the Neo4j graph database, that is so popular, has given more relevance to the work, as a suggestion it could be very good if the source code of the implementation would shared open-source with the community.

For what concerns Query 5, the search results are sorted by the path lenght. As a future extension for the language, I think it could be interesting to introduce a ranking mechanism that sort first certain sensors with respect to others, not only considering the path lenght. Also due the complexity of search inside a graph network it could be interesting the authors share the running time of the search queries.

Overall I think this is a very nice work, the Internet of water metafora is very powerfull. What has missed, is the evaluation of at least one between the efficiency and the effectiveness of the proposed language. At least the efficiency in term of memory consumption and cpu time should be discusses, and when is possible to be compared with a baseline system.

Author Response

Comment 1:  Several recent references approach the subject of this paper: the authors should 

refer them to substantiate their thesis. In the following I suggest several references that are related, 

it is up to the authors to check which are the most useful to better clarify the context of the paper and 

support their thesis.  When the authors say in rows 13-16: “There are many applications for sensor networks, 

from monitoring a single home, to the surveillance of a large city, to earthquake detection for the whole world. 

[ref1] For example, researchers, farmers, and governments need to monitor aspects of the natural environment 

such as air pollution, water quality, soil conditions, and weather metrics” [ref2]

Several recent references this issue. Among others:

[ref1] Cozza V., Guagliardi I., Rubino M., Cozza R., Martello A., Picelli M., Zhupa E., 2015. Esopo: 

Sensors and social pollution measurements. CEUR Workshop Proceedings, 1478, 52-57.

Ma, Yajie, Guo, Yike, Ghanem, Moustafa: Distributed pattern recognition for air quality analisys in sensor network system. 

IADIS International Journal on Computer Science and Information Systems Vol. 5, No.1, pp. 87-100ISSN:1646-3692

[ref2] Buttafuoco G., Guagliardi I., Tarvainen T., Jarva J., 2017. A multivariate approach to study the 

geochemistry of urban topsoil in the city of Tampere, Finland. Journal of Geochemical Exploration 181, 191-20

- Zuzolo D., Cicchella D., Lima A., Guagliardi I., Cerino P., Pizzolante A., Thiombane M., De Vivo B., Albanese S., 2020. 

Potentially toxic elements in soils of Campania region (Southern Italy): Combining raw and compositional data. 

Journal of Geochemical Exploration, 213, Article number 106524

 Reply 1:Thank you for these additional references. We   found the ones related for soil applications 

 relevant for the present paper, and included them in Section 1 (Introduction). 

Comment 2:  The authors say: “The core of the proposed query mechanismis a logic-based3 language 

that is capable to return time, value and time series outputs, as well as Boolean queries”. This issue has 

been investigated in several papers that take into account spatial-temporal queries, in line with the 

Jensen work already mentioned by the authors as [11], there are also:

Cozza, V., Messina, A., Montesi, D., Arietta, L., Magnani, M.: Spatio-temporal keyword queries in 

Social Networks. In Springer, editor, B. Catania, G. Guerrini, and J. Pokorn(Eds.): ADBIS 2013, volume 8133 of LNCS, pages 70-83, 2013.

Anand, A., Bedathur, S., Berberich, K., Schenkel, R.: Efficient Temporal Keyword 

Queries over Versioned Text. In: Proc. of ACM CIKM Conf. (2010)

Berberich, K., Bedathur, S., Neumann, T., Weikum, G.: A time machine for text search. 

In: Proceedings of the 30th Annual International ACM SIGIR Conference on Research and 

Development in Information Retrieval - SIGIR 2007, p. 519 (2007)

 Reply 2:Thank you for these resources. We found the first one appropriate and commented on it. 

 We believe that the works by Anand et al and 

 Berberich et al., although related with the topic, fall somehow out of the scope of the paper, and 

 decided to skip them, although we thank the reviewer for the suggestion.

Comment 3:  Moreover, It seems the submitted paper is a follow up work of [32], 

however [32] is not yet published so I cannot read on it. What is needed is that the authors 

at least mention how the present paper differs from that one.

 Reply 3: While this work was under review, that paper (now reference [34]) has been published online and 

 can be accessed via the DOI in the reference. We have also  included a short discussion in this paper, 

 indicating the differences, specifically pointing out that the previous work is focused on topological issues 

 (and a comparison of relational and graph databases), and does not address time-series data at all.

Comment 4:  This idea presented in the paper is very interesting, and the proof-of-concept implementation 

over the Neo4j graph database, that is so popular, has given more relevance to the work, as a suggestion 

it could be very good if the source code of the implementation would shared open-source with the community. 

 Reply 4:Thank you for this positive comment. We can of course share the code, even though it is an 

 initial implementation and many improvements are possible. The data and the code will be made 

 available using the OSF platform with DOI 10.17605/OSF.IO/95TCX.

Comment 5:  For what concerns Query 5, the search results are sorted by the path length. As a future 

extension for the language, I think it could be interesting to introduce a ranking mechanism that sort first 

certain sensors with respect to others, not only considering the path length. Also due the complexity of 

search inside a graph network it could be interesting the authors share the running time of the search queries.

Overall I think this is a very nice work, the Internet of water metafora is very powerfull. What has missed, is 

the evaluation of at least one between the efficiency and the effectiveness of the proposed language. 

At least the efficiency in term of memory consumption and cpu time should be discusses, 

and when is possible to be compared with a baseline system.

 Reply 5:Thank you for these additional suggestions. We added a discussion on the implementation 

 and limitation of the representation used, as well as examples of the limitation of the proposed theoretical approach, 

 in Section 6 (Discussion and future work) and partially in Section  5.2 (Data model and Cypher extensions). 

 With respect to the results of the queries executed, we have added a small discussion and an idea of the execution times, 

 but we have not further elaborated on this topic, because we wanted to focus on the model and proposed language, and 

 not on the Neo4j implementation. Addressing execution times is not the main point of this paper. 

 Anyway, we will of course  consider these issued¿s for future extensions of this work.  

Round 2

Reviewer 1 Report

The authors have adequately responded to all the review comments / suggestions and is therefore, in my opinion, ready for publication.